# Forward genetic screening using fundus spot scale identifies an essential role for *Lipe* in murine retinal homeostasis

Seher Yuksel [1,4], Bogale Aredo[1,4], Yeshumenesh Zegeye [1], Cynthia X. Zhao[1], Miao Tang[2], Xiaohong Li[2], John D. Hulleman [1], Laurent Gautron [3], Sara Ludwig[2], Eva M. Y. Moresco [2], Igor A. Butovich [1✉], Bruce A. Beutler [2✉] & Rafael L. Ufret-Vincenty [1✉]

Microglia play a role in the pathogenesis of many retinal diseases. Fundus spots in mice often correlate with the accumulation of activated subretinal microglia. Here we use a semi-quantitative fundus spot scoring scale in combination with an unbiased, state-of-the-science forward genetics pipeline to identify causative associations between chemically induced mutations and fundus spot phenotypes. Among several associations, we focus on a missense mutation in *Lipe* linked to an increase in yellow fundus spots in C57BL/6J mice. *Lipe*$^{-/-}$ mice generated using CRISPR-Cas9 technology are found to develop accumulation of subretinal microglia, a retinal degeneration with decreased visual function, and an abnormal retinal lipid profile. We establish an indispensable role of *Lipe* in retinal/RPE lipid homeostasis and retinal health. Further studies using this new model will be aimed at determining how lipid dysregulation results in the activation of subretinal microglia and whether these microglia also play a role in the subsequent retinal degeneration.

[1] Department of Ophthalmology, UT Southwestern Medical Center, Dallas, TX, USA. [2] Center for the Genetics of Host Defense, University of Texas Southwestern Medical Center, Dallas, TX, USA. [3] Center for Hypothalamic Research and Department of Internal Medicine, University of Texas Southwestern Medical Center, Dallas, TX, USA. [4] These authors contributed equally: Seher Yuksel, Bogale Aredo. ✉email: Igor.Butovich@UTSouthwestern.edu; Bruce.Beutler@UTSouthwestern.edu; Rafael.Ufret-Vincenty@UTSouthwestern.edu

A large number of studies have shown an association between activated retinal microglia and diseases such as age-related macular degeneration (AMD), retinal dystrophies and diabetic retinopathy[1–9]. There is still debate on whether these cells are deleterious (by promoting inflammation)[9–15] or homeostatic (by removing debris)[16]. In fact, it is possible that under different circumstances they may be either[7,10,17–20]. We propose that identifying genes that modulate the accumulation of subretinal microglia will help us identify pathways that are essential for retinal homeostasis. Furthermore, careful characterization of the resulting mouse models may also shed some light on the mechanisms regulating immune surveillance, cell trafficking, and neuroinflammation in the retina. With these goals in mind, here we apply a fundus spot scale screen to our state-of-the-science forward genetics pipeline[21,22], whose greatest advantage compared to other forward genetics protocols[23–29], is the fact that all mice screened are G3 mice that have been pre-genotyped at all mutant loci, allowing for prompt determination of causative mutations[22]. This affords us the opportunity to pursue gene-phenotype associations that others may ignore. In traditional forward genetic screens, there is a huge amount of effort involved in pursuing any potential "hits" because many of them utilize outcross/intercross to a mapping strain in combination with exome sequencing to retrospectively identify causative mutations, a process that is time-consuming. This discourages the pursuit of any associations that do not lead to severe pathology.

Although the exact composition of the white/yellow subretinal spots seen in mouse fundus photos in various models is not entirely certain, we and others have previously shown they correlate well with Iba1+ subretinal microglia[10,30–32]. We developed a scale[33], to perform rapid and reproducible semiquantitative analysis of these spots. Here, we used this fundus spot scale to screen close to 6000 G3 mice for mutations that altered fundus spot scores compared to wild-type mice. We identified several genes that, when mutated, led to fundus spot accumulation. Six of these genes are already known to cause retinal pathology (Supplementary Table S1), providing proof of principle. From the gene-phenotype associations, we decided to pursue the *Lipe* gene, both because its mutation led to a strong phenotype of fundus spot accumulation and because the fundus spot scale was the only screening parameter that allowed us to identify it.

In this work we also demonstrate that a CRISPR-generated $Lipe^{-/-}$ mouse line reproduces our findings of extensive accumulation of fundus spots in retinal photos. We show that they develop accumulation of subretinal microglia and a retinal degeneration. We then characterize these mice with emphasis on retinal imaging, retinal anatomy and ultrastructure, in situ hybridization, electrophysiology, and immunohistochemistry. Finally, using ultra high-performance liquid chromatography with mass spectrometry, we demonstrate that the $Lipe^{-/-}$ mice display an anomaly in lipid metabolism in the retina and RPE.

## Results

**Application of a fundus spot semiquantitative scale as a screening tool on a forward genetics pipeline is useful in identifying genes important to retinal homeostasis**. With the goal of identifying genes essential to retinal development and homeostasis, and in particular genes with an impact on retinal immune cell activation, we developed[34] and applied a semi-quantitative fundus spot scoring scale to fundus photographs obtained from 4 to 6-month-old third generation (G3) mice carrying heterozygous and homozygous mutations induced by N-ethyl-N-nitrosourea (ENU) in their G0 ancestors[21,22](Fig. 1a–c). After screening over 5906 mice from the ENU-generated pedigrees, we have achieved a 4.72% saturation of the genome when

considering genes with probable null or damaging mutations, and at least 2 mice in the homozygous state. As a proof of concept, the fundus spot scale identified 6 genes with known association to retinal degeneration during our screening experiments (see Supplementary Table S1). It also identified additional genes with a potential association to fundus spot accumulation that had not been previously described. One of these associations involved the *reservoir* allele, encoding a missense mutation in the *Lipe* gene that caused a serine to a proline substitution at amino acid 743 of the enzyme hormone sensitive lipase E (HSL). On further analysis, the *reservoir* allele had several noteworthy characteristics: (1) It showed a single peak in the Manhattan plot (Fig. 1g). Also, in addition to an absence of confounding mutations that could be responsible for the findings, the *p* value for the association of the reservoir allele to fundus spots was $3.5 \times 10^{-15}$ (Fig. 1h). Of note, mice heterozygous for the *reservoir* allele did not show an increased accumulation of fundus spots, indicating the autosomal recessive inheritance of this trait. (2) There were 2 VAR mice (homozygous for the mutation of interest) in the pedigree, and they both showed a similar and severe phenotype (many fundus spots). (3) No other phenotypes were detected by OCT in G3 mice homozygous for the reservoir allele, suggesting that fundus spot accumulation was the strongest phenotype (Fig. 1i). (4) The Polyphen2 software predicted the mutation to be probably damaging, while the gene was not predicted to be essential to survival, and (5) The Candidate Explorer software predicted the gene to be a potential candidate[35]. Based on these findings we established that applying a semiquantitative fundus spot scale as a screening tool to our forward genetics pipeline, and using continuous variable phenotype data for linkage analysis, is an effective approach to identify genetic mutations associated to both the accumulation of fundus spots and the development of retinal degeneration. Moreover, we chose *Lipe*, identified solely by the fundus spot semiquantitative scale screen, as an interesting gene for further studies.

**A CRISPR-generated $Lipe^{-/-}$ mouse line reproduced the *reservoir* phenotype**. Since each mouse in our ENU mutagenesis screen contains over 60 different mutations, our first goal was to confirm that *Lipe* was indeed the gene responsible for the *reservoir* phenotype, since it was seen in just 2 homozygous mice within a single pedigree. To this end we generated $Lipe^{-/-}$ mice using CRISPR-Cas9 mediated gene targeting. For our $Lipe^{-/-}$ mouse line, from several mutations seen in the founder mice, we chose one that led to a 14 bp deletion. It resulted in an early termination of the protein product at amino acid 111 (normal length is 802 amino acids; Supplementary Fig. S1a–d). We maintained breeding stocks of the CRISPR mice to generate different age groups of both the $Lipe^{-/-}$ and $Lipe^{+/+}$ colonies. We then proceeded to obtain fundus photographs of $Lipe^{-/-}$ vs $Lipe^{+/+}$ mice at several ages from 2 to 13 months. Two investigators blinded to genotype independently scored each fundus image according to the semiquantitative scale. We first determined that there was no gender-related difference in the accumulation of fundus spots in $Lipe^{-/-}$ mice (Supplementary Fig. S2). However, we did observe a striking accumulation of fundus spots in $Lipe^{-/-}$ mice compared to $Lipe^{+/+}$, confirming the expected phenotype (see Fig. 2). The difference was already statistically significant at 3 months of age, as seen in the CRISPR $Lipe^{-/-}$ founder mice (see Supplementary Fig. S3). By 6–8 months of age, about 52% of mice had reached a score of 6 or more. The difference remained highly significant at all age groups until at least 13 months of age. Despite a high fundus spot score being achieved at an early age in $Lipe^{-/-}$ mice, a linear regression analysis suggested that these spots continued to show a moderate increase with age ($R^2 = 0.235$; $F = 26.1$, $p = 1.96 \times 10^{-6}$).

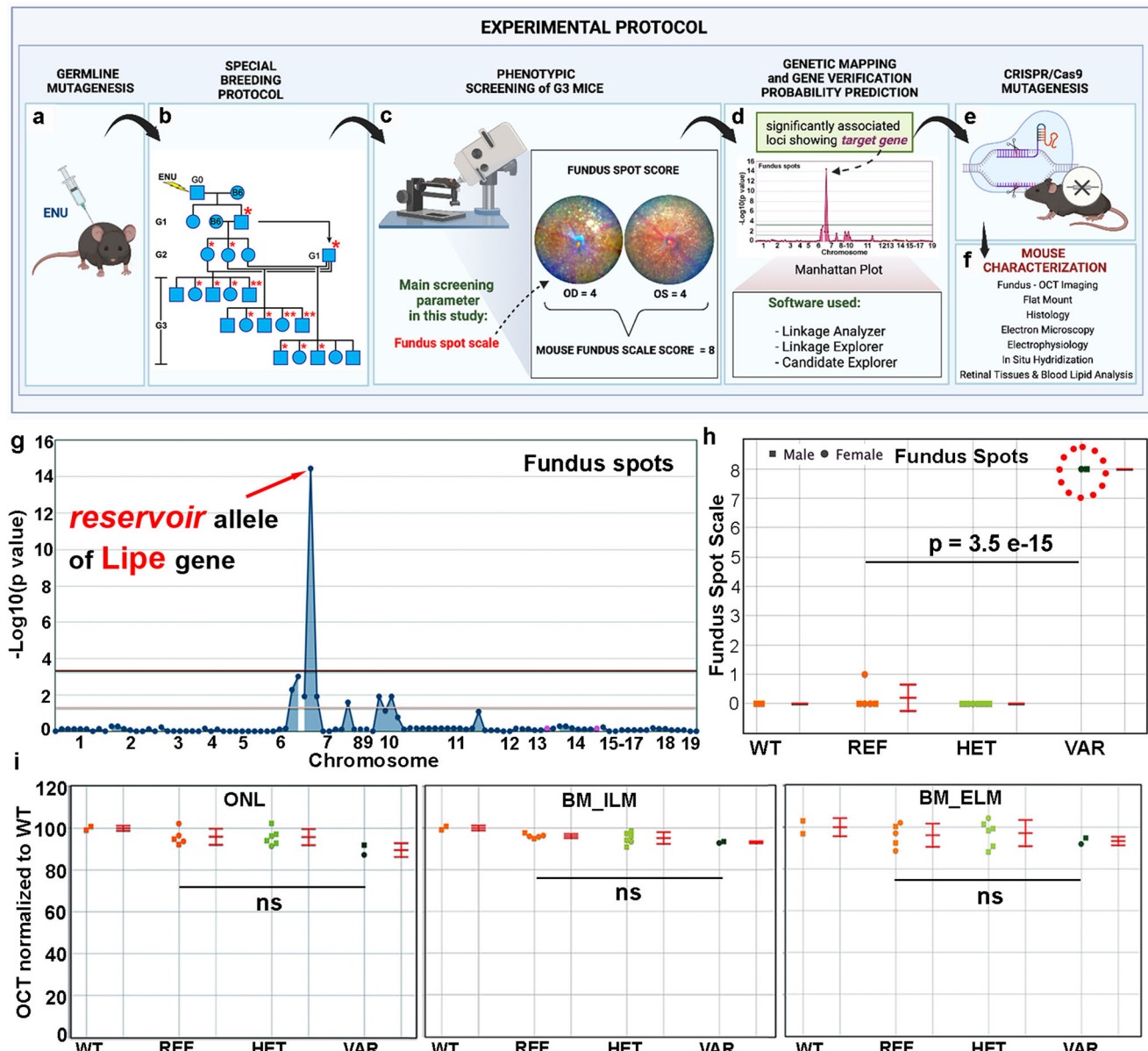

**Fig. 1 Experimental protocol for forward genetics screening, identification of gene-phenotype associations to fundus spot accumulation and characterization of phenotypes.** Mutagenesis was induced with N-ethyl-N-nitrosourea (ENU) (**a**). A special breeding protocol is used to ensure that we include potential dominant and recessive mutations (**b**). G3 mice were screened using a fundus spot semiquantitative scale (**c**). Statistically significant gene-phenotype associations were identified using special software (**d**). Genes of interest were targeted using CRISPR/Cas9 technology (**e**) and the resulting mice were phenotyped (morphology, ultrastructure, physiology, biochemistry) with a variety of techniques (**f**). Using this protocol, the reservoir allele was identified, and a Manhattan Plot (**g**) and scatter plot (**h**) show its statistically significant association to fundus spot accumulation. No significant association between the reservoir allele and retinal layer thickness was detected in this pedigree for any of our OCT parameters (**i**). The scatter plots (**h**, **i**) show individual mice plus the error bars (Mean ± Standard Deviation). Figure 1a–f was generated with BioRender.com.

Of note, we also found a myriad of hyperautofluorescent spots when $Lipe^{-/-}$ mice were imaged using fundus autofluorescence (FAF). Although it is difficult to establish if the autofluorescent spots correspond to the fundus spots, images of eyes in which the distribution of fundus spots is asymmetric and matches the distribution of autofluorescent spots (see Supplementary Fig. S4) would suggest that this is the case.

**$Lipe^{-/-}$ mice have an increased accumulation of subretinal microglia.** Prior studies from our group and others[10,30–32,34,36] in multiple mouse models strongly suggest that fundus yellow spots correlate with the presence of subretinal Iba1+ cells. To

determine if the fundus spots in $Lipe^{-/-}$ mice were also associated with increased subretinal microglia, we prepared RPE-choroid-sclera flat mounts (referred to as "RPE flat mounts" below) and stained them for Iba1 (Fig. 3). All RPE flat mount samples were imaged at 20× magnification as described in the Methods section (Fig. 3a). In brief, one image from each antero-posterior region (i.e. central, paracentral, midperipheral and peripheral) of the flat mount was obtained in each quadrant. Iba1+ cells were counted by a masked investigator. A statistically significant elevation in the Iba1+ cell count was seen in all flat mount regions of $Lipe^{-/-}$ mice compared to $Lipe^{+/+}$ (Fig. 3b). Applying the interquartile range (IQR) method (see Statistical Analysis sub-section in Methods), we identified one 20× field in

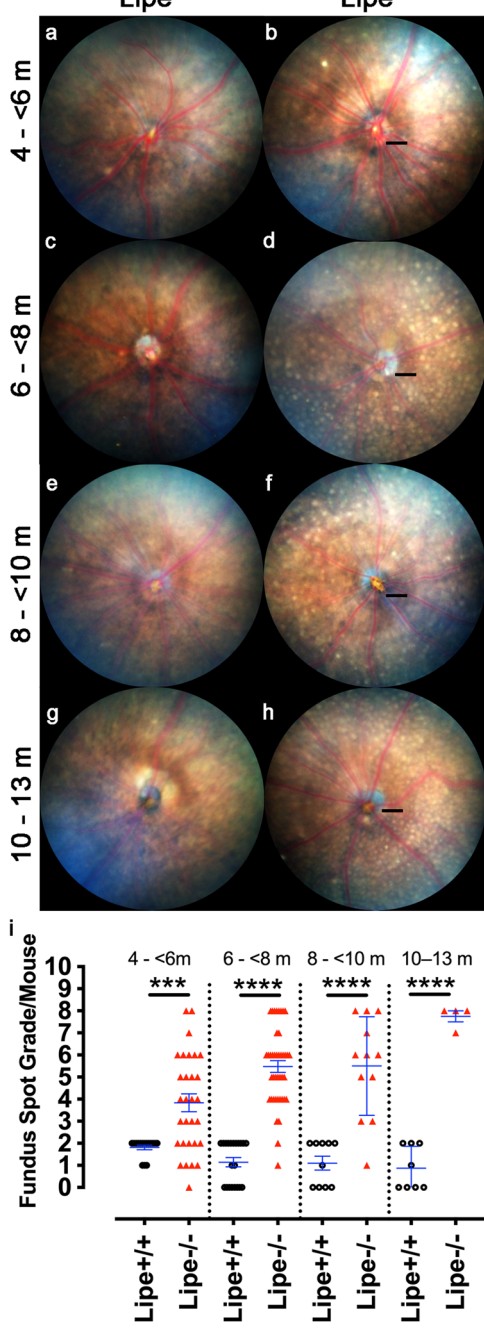

**Fig. 2 Clinical course of fundus spot accumulation in *Lipe*⁻/⁻ mice compared to *Lipe*⁺/⁺.** At 4-6 months of age, *Lipe*⁻/⁻ mice show a clear increase in fundus spot accumulation compared to *Lipe*⁺/⁺ mice (**b** vs. **a**). As the mice age there is a progressive increase in fundus spots (**c–h**). The difference was statistically significant for all age groups (**i**). The experiment included *Lipe*⁻/⁻ (n = 30, 42, 12 and 4 mice for age groups 4-<6, 8-<8, 8-<10, and 10-<13 months, respectively) and *Lipe*⁺/⁺ (n = 16, 21, 10 and 8 mice for age groups 4-<6, 8-<8, 8-<10, and 10-<13 months, respectively). Black scale bars = 200 μm. Data are shown as Mean ± SEM. Two-tailed Student's *t* test. **p < 0.01, ***p < 0.001, ****p < 0.0001.

the peripheral region, and two in the midperipheral region that were outliers. When eliminating those from the statistical analysis, the significance became even higher (p < 0.01, blue ellipses in Fig. 4b). We then stained flat mounts for both Iba1 and CD16 (Fig. 3c–e), since CD16 is a marker of microglial activation[10]. As expected, microglia showing morphological evidence of activation

(less extensions and a larger and rounder cell body; white squares in Fig. 4d) were the ones showing stronger co-staining for CD16 and Iba1 (yellow color in Fig. 3c, d). Interestingly, we found that a large proportion of the subretinal microglia were CD16+ (Fig. 3e). Furthermore, the percentage of CD16+ microglia was significantly larger in the central region compared to the paracentral region (58 ± 6% vs. 40 ± 4%, p = 0.033).

To better understand the microglial cells in *Lipe* deficient mice, we also prepared retinal flat mounts plus additional RPE flat mounts and imaged them with confocal microscopy after immunohistochemistry. Interestingly, retinal flat mounts demonstrated that Iba1+ cells seen in the inner plexiform layer and outer plexiform layer of the retina have a similar morphology and distribution in *Lipe*⁻/⁻ mice compared to *Lipe*⁺/⁺ (Fig. 4b vs a, and e vs d). These cells do not stain for CD16 (Fig. 4c, f). Meanwhile, cells in the subretinal space (stuck to the surface of RPE cells in the RPE flat mounts) are seen in large numbers in *Lipe*⁻/⁻ (Fig. 4h, k) eyes, but rarely in *Lipe*⁺/⁺ eyes (Fig. 4g, j). These cells also stain for CD16 (Fig. 4i, l). Moreover, both the retinal and the subretinal immune cells are TMEM119+ suggesting that they are microglia rather than infiltrating monocyte/macrophages (Supplementary Fig. S5). In fact, RPE flat mounts stained for F4/80 and TMEM119 (Supplementary Fig. S6) showed that the F4/80-stained cells (F4/80 recognizes both microglia and macrophages) are also TMEM119+ (specific marker for microglia), but CCR2 negative (CCR2 is a marker for infiltrating macrophages).

Of note, superimposing the fundus photographs over the infrared images from the Spectralis OCT, using the vasculature as guidance to match the images, we were able to find that the yellow spots on fundus photographs correlated to tiny hyperreflective spots just above the RPE/interdigitation zone (RPE/IZ) on OCT (Supplementary Fig. S7a). Furthermore, using the registration feature of the Spectralis, we could monitor some of these with time, and found that sometimes new spots would appear or disappear from a particular location (Supplementary Fig. S7b). To better define the nature of these changes, we imaged three different *Lipe*⁻/⁻ eyes on 3 separate sessions, 2 wks apart (Supplementary Fig. S7c). Reviewing these images, it is clear that within a 2 wk span of time some new spots appear, some disappear, and many of them moved from their original location.

In order to ensure that the fundus spots did not represent clumps of dying cells, TUNEL staining of retinal sections was performed. It did not show any differences when comparing *Lipe*⁺/⁺ and *Lipe*⁻/⁻ mice (Supplementary Fig. S8). Both of these mouse lines had only occasional isolated apoptotic cells. There was no evidence of clumping of apoptotic cells, and their number and size would not account for the fundus spots. Furthermore, staining of retinal sections for cone arrestin does not show any clumping of cones in *Lipe*⁻/⁻ mice compared to wild type (Supplementary Fig. S9).

Interestingly, in one *Lipe*⁻/⁻ mouse we observed a lesion on OCT that appeared suggestive of choroidal neovascular activity (see Supplementary Fig. S10a–f). This was seen when the mouse was 10 months old (Fig. 10a–c), but not 3 months prior (Fig. 10d–f). However, this was the only mouse for which we observed this issue, so its significance is unclear. Of note, we only obtained large cube OCT scans in 6 *Lipe*⁻/⁻ and 4 *Lipe*⁺/⁺ mice that were imaged with the Spectralis OCT in order to use the registration capabilities of that instrument.

Given that myopic degeneration can lead to lacquer cracks and occasionally choroidal neovascularization, we wanted to rule out the possibility that the fundus spots in *Lipe*⁻/⁻ mice could represent similar breaks in RPE/Bruch's membrane/choriocapillaris due to increased eye length. Anteroposterior measurements of freshly enucleated eyes using calipers demonstrated that the

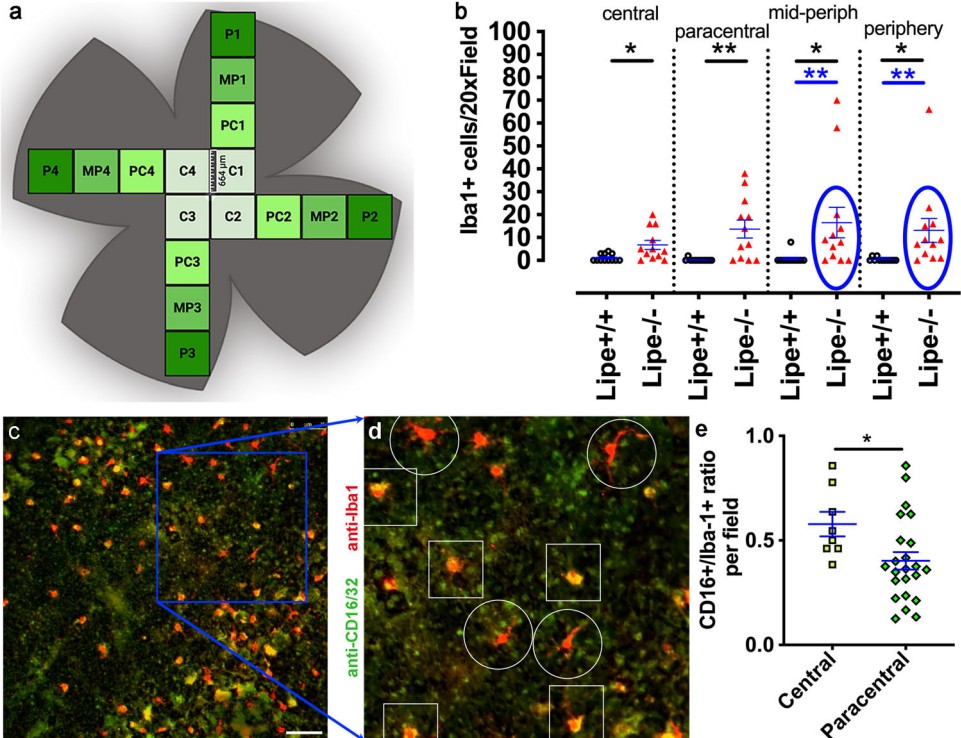

**Fig. 3 *Lipe*$^{-/-}$ mice demonstrate increased accumulation of activated subretinal microglia. a** Diagram showing the imaging protocol for RPE flat mounts with each square representing a 20× field (664 μm by 664 μm). Central (C), paracentral (PC), mid-peripheral (MP) and peripheral (P) regions were imaged in each quadrant. This diagram was generated with BioRender.com. **b** Iba1+ cells were counted in flat mounts from *Lipe*$^{-/-}$ and *Lipe*$^{+/+}$ mice (*n* = 12 fields per group per region). *Lipe*$^{-/-}$ mice showed a statistically significant increase in Iba1+ cells in all imaged regions (central, paracentral, mid-peripheral and peripheral) compared to *Lipe*$^{+/+}$ mice. Data are presented as Mean ± SEM. Two-tailed Student's *t* test; *$p < 0.05$, **$p < 0.01$. Analysis was done both including all values (black asterisks), and also excluding outliers (blue ellipses and blue asterisks). **c** Flat mounts were then co-stained with Iba1 (red) and CD16 (green) and a representative 20× image is shown. **d** A magnified view of the blue squared area in Fig. 3c is shown, to highlight examples of microglia with typical activation morphology (less extensions and larger cell bodies; shown in white squares). Note that microglia with morphological signs of activation are also co-staining with CD16 and are thus appearing more orange/yellow. These should be compared to those in white circles, which have less of an activation morphology (more extensions) and are not staining for CD16 (they appear more red). Scale bar in panel **c** = 75 μm. **e** CD16+/Iba1+ ratio obtained from central (*n* = 8) and paracentral (*n* = 23) fields. The central fields show a statistically significant increase in the proportion of Iba1+ cells that are also CD16+ when compared to paracentral fields in *Lipe*$^{-/-}$ flatmounts. Data are presented as Mean ± SEM. Two-tailed Student's *t* test. *$p < 0.05$, **$p < 0.01$.

eyes of *Lipe*$^{-/-}$ mice were of the same size as *Lipe*$^{+/+}$ eyes (see Supplementary Fig. S10g).

Our findings suggest that yellow spots seen on fundus exam in *Lipe*$^{-/-}$ mice are associated with subretinal Iba1+/CD16+/TMEM119+/CCR2− cells consistent with activated microglia, and that these can be seen as tiny hyperreflective spots over the RPE/IZ on OCT. We hypothesize that the lack of *Lipe* expression in the retina leads to activation of retinal microglia and migration into the subretinal space.

**Progressive outer retinal thinning is observed in *Lipe*$^{-/-}$ mice with aging.** To investigate whether retinal structural changes also develop in *Lipe*$^{-/-}$ mice, we used both optical coherence tomography (OCT) to characterize changes over time and standard histology. OCT images were collected from *Lipe*$^{-/-}$ and *Lipe*$^{+/+}$ mice at different ages (from 2 to 13 months old) using a Phoenix MICRON OCT2 system. Masked investigators then used Image J to measure several parameters from these images (total retinal thickness, outer retinal thickness, and outer nuclear layer: TRT, ORT and ONL, respectively; see methods). First, we compared measurements from male *Lipe*$^{-/-}$ vs female *Lipe*$^{-/-}$ mice and were able to determine that there were no gender-specific differences (Supplementary Fig. S11), so further analyses were done

combining males and females. Qualitative inspection of the images showed some outer retinal disruption; by 3 months of age, we could see a decrease in the thickness and demarcation of the normally hypo-reflective photoreceptor outer segment band under the ellipsoid zone (Fig. 5b vs a and d vs c. We also found that the retina was thinner in *Lipe*$^{-/-}$ mice compared to age-matched *Lipe*$^{+/+}$ mice (Fig. 5d vs c, OCT). In fact, TRT, ONL and ORT were all decreased in *Lipe*$^{-/-}$ mice beginning at 4 months of age (Fig. 5e, f and g, respectively). The thinning showed a moderate progression with age (linear regression for ONL shows: $R^2 = 0.372$; $F = 25.46$, $p = 8.71 \times 10^{-6}$). While the retinal thinning was much less dramatic than the one we had documented in mice with an Sfxn3 deficiency[21] (e.g. close to 20% decrease in ONL at 1 year in *Lipe*$^{-/-}$, vs 80% in *Sfxn3*$^{-/-}$ mice), it was still highly significant ($p < 0.001$ for most comparisons). Furthermore, we found that the retinal thinning mostly involved the outer retina; no decrease in the ganglion cell complex thickness (GCC; see methods for definition) was noticed even at older ages (Fig. 5h).

We also prepared histological specimens from 10-month-old *Lipe*$^{-/-}$ and *Lipe*$^{+/+}$ mice using a freeze-substitution technique (Fig. 6a, b). H&E-stained retinal sections were imaged at 20× magnification sequentially across the entire section for both *Lipe*$^{-/-}$ and *Lipe*$^{+/+}$ mouse eyes. A masked investigator first

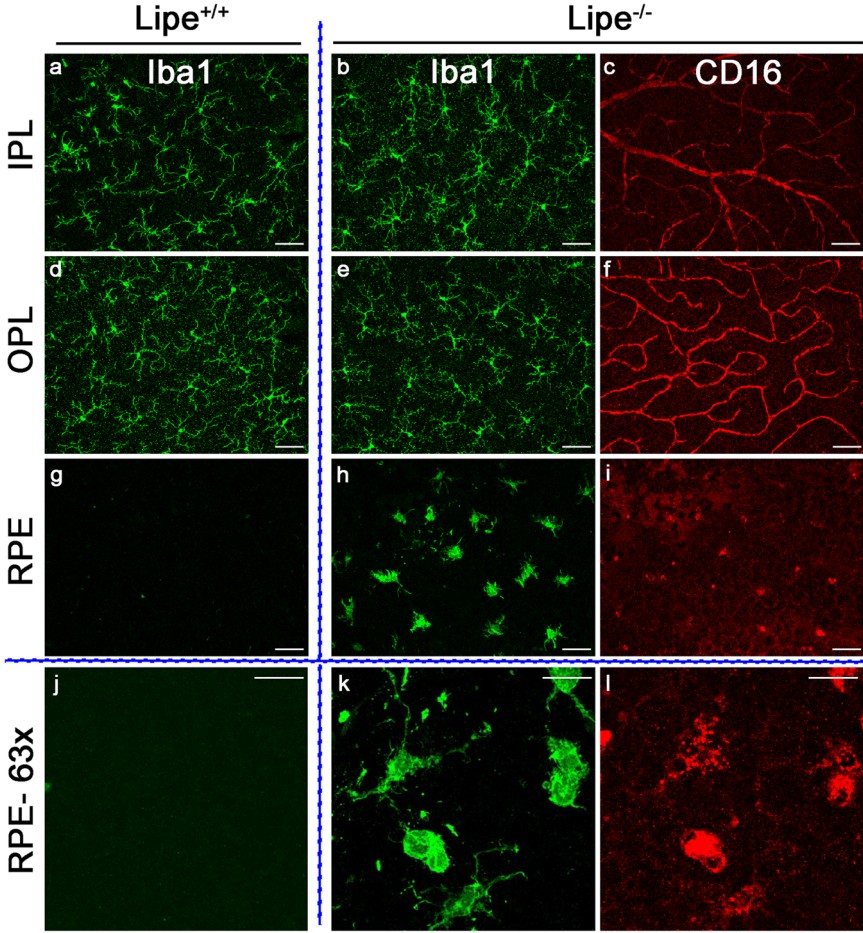

**Fig. 4 *Lipe* deficient mice have similar Iba1+ cells in the retina as control mice but accumulate CD16+ subretinal microglia with activated morphology.** Retina (**a–f**) and RPE (**g–l**) flat mounts from *Lipe*$^{+/+}$ (**a, d, g, j**) and *Lipe*$^{-/-}$ (**b, c, e, f, h, i, k, l**) mice were stained using anti-Iba1 and anti-CD16 antibodies and imaged using confocal microscopy. For the retina flat mounts, the inner plexiform layer (IPL) and outer plexiform layer (OPL) were imaged and showed many Iba1+ cells with small cell bodies and long branching extensions, which were similar in the *Lipe*$^{+/+}$ (**a, d**) and *Lipe*$^{-/-}$ (**b, e**) mice. These cells did not show CD16 staining. However, while the RPE flat mounts of *Lipe*$^{+/+}$ were for the most part devoid of Iba1+ cells (**g, j**), there were many Iba1+ cells in *Lipe*$^{-/-}$ eyes (**h, k**). These cells stained for CD16 and had a very different morphology compared to those in the retina, with larger cell bodies and a lower number of extensions which were also shorter, suggesting an activated state. Scale bars in **a–i** = 50 μm. Scale bars in **j–l** = 25 μm.

counted ONL nuclei at 300 μm intervals. We were able to demonstrate a significant decrease in ONL nuclei in *Lipe*$^{-/-}$ mice (Fig. 6c). We then used image J to measure ONL thickness at the same locations. Statistically significant reductions in ONL thickness were again seen in *Lipe*$^{-/-}$ mice compared to WT (Fig. 6d). While anatomical and structural preservation is optimal in freeze-substitution specimens, we confirmed this finding of decreased ONL thickness using cryo specimens that do not involve potential lipid extraction solvents (Supplementary Fig. S12).

We conclude that disruption of the *Lipe* gene led to a progressive photoreceptor degeneration. This is a strong indication that expression of its protein product HSL is needed in order to maintain retinal integrity, particularly in the outer retina.

**_Lipe_ is expressed in both retina and the RPE.** Having documented a retinal degeneration in mice with a global deficiency of *Lipe*, we decided to test whether HSL is normally expressed in the retina and/or RPE-choroid-sclera of WT C57BL/6J mice. HSL has several isoforms, which are generated by use of alternative translational start codons, and which are expressed in different tissues[37,38]. The most common of these are a long isoform (120–130 kDa) and a short isoform (80–90 kDa), both of which

are expressed in testes. Thus, we used testes as a positive control for our assay. We isolated posterior eye cups of C57BL/6J mice and separated them into retina vs. RPE-choroid-sclera. After protein isolation and concentration of the samples, Western blot analysis revealed expression of the short isoform of HSL in the retina (Fig. 7a, lanes 2,3; 30 μg loaded) of WT mice. The RPE (lanes 4,5; only 20 μg loaded for the RPE) also showed HSL expression, and it seemed to be slightly higher than in the retina. In both the retina and RPE the two predicted bands for the short HSL isoform (81 and 83 kDa) were detected. We did not detect the long isoform. Of note, both of these ocular tissues seemed to have an expression of HSL per μg of tissue that was lower than that in testes (lane 1, 40 μg loaded), which is known to have very high expression of HSL. Neither one of the HSL isoforms were detected in the retina, RPE or testes of *Lipe*$^{-/-}$ mice (lanes 6, 7 and 8, respectively; all had 40 μg loaded). We used both beta-actin and GAPDH antibodies as loading controls. Of interest, the beta-actin seems to have higher expression in *Lipe*$^{-/-}$ tissues compared to *Lipe*$^{+/+}$. On the other hand, GAPDH has higher expression in retina than in other tissues, both for *Lipe*$^{-/-}$ and *Lipe*$^{+/+}$ samples.

In an effort to get more specific information about the location of expression of *Lipe* within the retina, we performed in situ

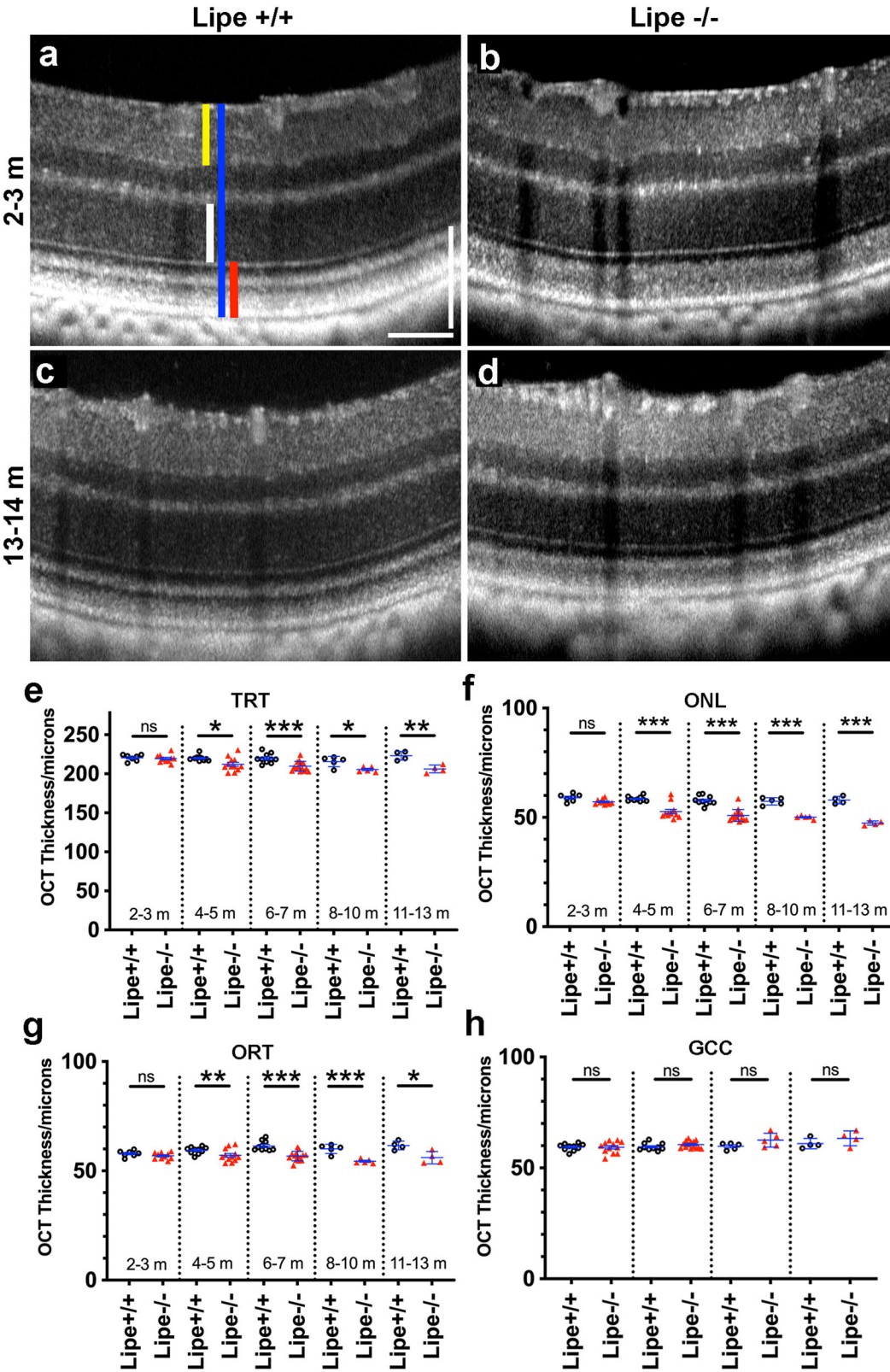

hybridization in retinal sections using RNAscope. Application of the *Lipe* probe to the WT mouse retina (Fig. 7b) led to a very different signal distribution compared to that seen using the *Sfxn3* probe (Fig. 7d). While *Sfxn3* mRNA appeared to be expressed predominantly in the inner nuclear layer and ganglion cell layer, *Lipe*'s signal was strongest in the outer nuclear layer, with some lower signal in the inner nuclear layer and perhaps the RPE.

Positive and negative controls for ISH using testis are shown in Fig. 7c and e, respectively.

To our knowledge, this is the first description of the expression and localization of *Lipe* in the mouse retina and RPE. Taken together, these findings confirmed that HSL is indeed expressed in the retina and also the RPE. Furthermore, it suggests that within the retina, photoreceptors account for the bulk of *Lipe* expression.

**Fig. 5 *Lipe*⁻/⁻ mice develop mild-moderate progressive outer retinal thinning with age.** OCT images were analyzed to measure total retinal thickness (TRT, blue line in panel **a**), outer nuclear layer (ONL, white line), outer retinal thickness (ORT, red line) and ganglion cell complex (GCC, yellow line). Representative OCT scans (**a**–**d**) demonstrate qualitative hyperreflective changes in the photoreceptor outer segments of *Lipe*⁻/⁻ mice from an early age (2–3 months old). ONL appears fairly normal in *Lipe*⁻/⁻ mice at that age, but shows some thinning by 13–14 months of age. Quantitative analysis (**e**–**h**) demonstrates that *Lipe*⁻/⁻ mice develop statistically significant thinning of TRT (**e**), ONL (**f**), and ORT (**g**), compared to age-matched *Lipe*⁺/⁺ mice. OCT thickness for figures **e**–**g** was quantified for *Lipe*⁺/⁺ ($n = 7, 10, 10, 5$, and 4 eyes at 2–3, 4–5, 6–7, 8–10, and 11–13 months old, respectively) and *Lipe*⁻/⁻ ($n = 10, 12, 14, 5$, and 4 eyes at 2–3, 4–5, 6–7, 8–10, and 11–13 months old, respectively). The thinning is statistically significant by 4–5 months of age and worsens progressively with aging. However, no thinning of the GCC was seen (**h**). OCT thickness for figure h was quantified for *Lipe*⁺/⁺ ($n = 10, 10, 5$ and 4 eyes at 4–5, 6–7, 8–10, and 11–13 months old, respectively) and *Lipe*⁻/⁻ mice ($n = 12, 14, 5$, and 4 eyes at 4–5, 6–7, 8–10, and 11–13 months old, respectively). Each data point represents an eye. One eye per mouse was analyzed. Data are presented as Mean ± SEM. Two-tailed Student's *t* test. *$p < 0.05$, **$p < 0.01$, ***$p < 0.001$.

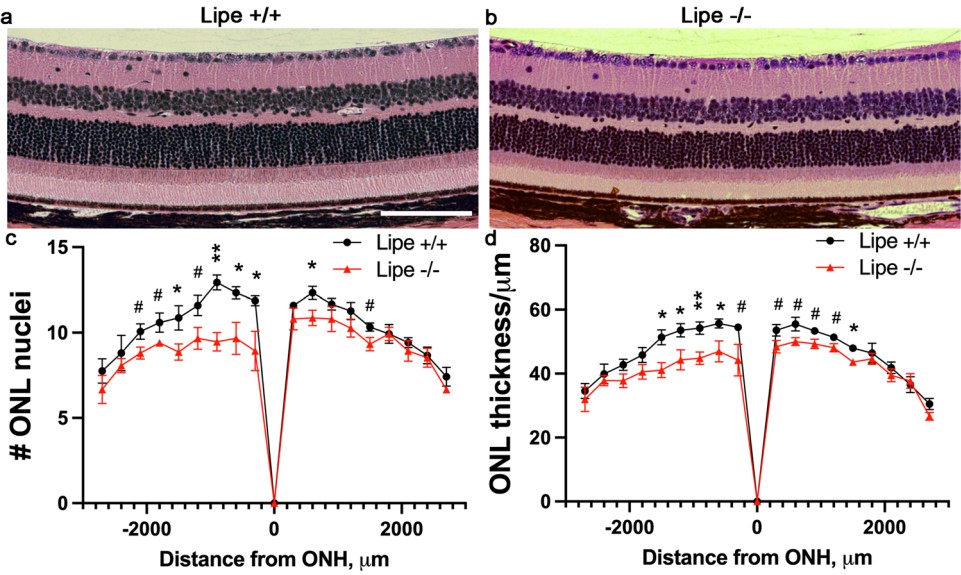

**Fig. 6 Analysis of histology sections shows thinning of the outer nuclear layer in *Lipe*⁻/⁻ mice.** Representative H&E images (20× magnification) of a *Lipe*⁺/⁺ control retina (**a**) and a *Lipe*⁻/⁻ retina (**b**) demonstrate moderate thinning of the ONL at 10 months of age. The number of nuclei in the ONL (**c**) and the thickness of the ONL (**d**) were measured on histology sections at 300 μm intervals starting from the optic nerve head (ONH). *Lipe*⁻/⁻ mice had a significantly thinner outer nuclear layer in comparison to *Lipe*⁺/⁺ controls based on both the nuclear count (**c**) and ONL thickness measurements (**d**). ONL counts and ONL thickness were obtained from *Lipe*⁺/⁺ ($n = 5$ eyes) and *Lipe*⁻/⁻ ($n = 5$ eyes). Data at each point of measurement are presented as Mean ± SEM. Two-tailed Student's *t* test. #$p < 0.1$, *$p < 0.05$; **$p < 0.01$. Scale bar = 100 μm.

***Lipe* deficiency leads to accumulation of degradation granule aggregates in the RPE.** Given the observed degenerative changes affecting the outer retina of *Lipe*⁻/⁻ mice, the accumulation of microglia right above the RPE, and the fact that *Lipe* is also expressed in the RPE of wild type mice, we decided to explore if we could detect any ultrastructural abnormalities in RPE cells of *Lipe*⁻/⁻ mice. Retinal degeneration often leads to increased waste and a resulting increase in lysosomal activity in the RPE[39–41]. Thus, we analyzed the melano-phago-lysosomal granule aggregates in RPE cells of *Lipe*⁻/⁻ vs WT mice. After obtaining transmission electron microscopy (TEM) images using a standard protocol (see Methods), a masked investigator counted all degradation granule aggregates (phagosomes, phagolysosomes, and melanolipofuscin; see Fig. 8a–f) in all TEM fields obtained (22–28 TEM fields per eye, 3 WT and 3 *Lipe*⁻/⁻ eyes). Figure 8g shows that the RPE cells of *Lipe*⁻/⁻ mice had a highly significant increase in granule aggregates per TEM field (4.6 vs 2.7, $p = 6.6 \times 10^{-8}$). The difference remained statistically significant when comparing the average TEM field count from the 3 *Lipe*⁻/⁻ eyes to that of the 3 WT eyes (Fig. 8h, $p = 0.008$). Of note, we did not observe any difference in RPE thickness or basal infolding thickness when comparing *Lipe*⁻/⁻ vs *Lipe*⁺/⁺ whether the comparison was done by mouse (Supplementary Fig. S13a, b) or by TEM field (Supplementary Fig. S13d, e). There was a mild

increase in Bruch's membrane thickness in *Lipe*⁻/⁻ that was significant when the comparison was made by TEM field (Supplementary Fig. S13c), but became only a trend when the comparison was made by mouse (Supplementary Fig. S13f, $p = 0.07$).

These data indicate that the absence of HSL leads to abnormalities in the lysosomal degradation processes within RPE cells. We speculate that this is due to increased degradation of photoreceptor outer segments, given the thinning in outer retinal parameters seen on histology and OCT. However, it is unclear if pathology within the RPE cells themselves could be partially responsible, since we showed that RPE cells also express HSL.

**Retinal function is reduced in aging *Lipe*⁻/⁻ mice.** Having demonstrated anatomic disturbances in the outer retina of *Lipe*⁻/⁻ mice, we decided to explore if these had functional consequences. We proceeded to evaluate full-field scotopic ERG responses of 10-month-old *Lipe*⁻/⁻ and *Lipe*⁺/⁺ mice after 12-h of dark adaptation (Fig. 9). A comparison of the normalized a-wave and b-wave signals of *Lipe*⁻/⁻ vs. *Lipe*⁺/⁺ mice was performed for both a low intensity (0.1 log cd.s.m⁻²) and a moderate intensity (1 log cd.s.m⁻²) stimulus. The results show a significant reduction of the a-wave in *Lipe*⁻/⁻ mice with both the

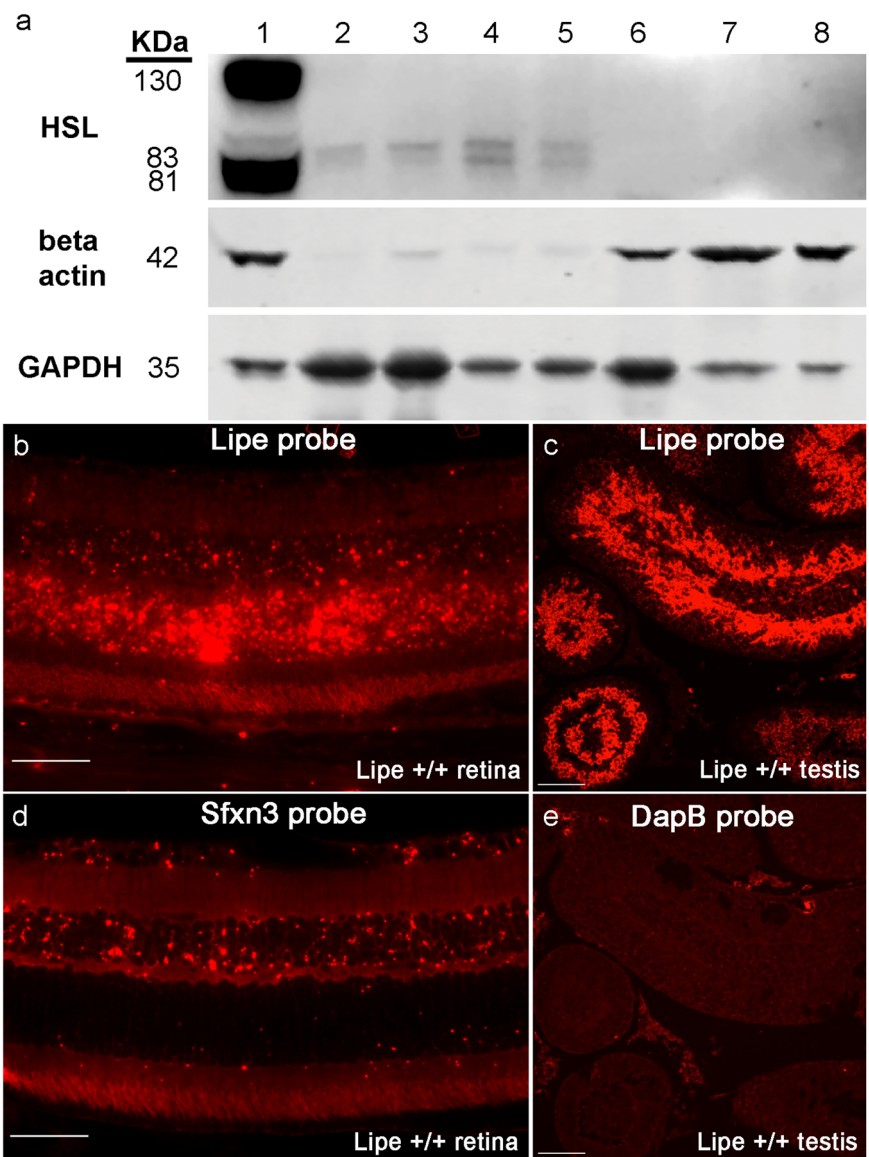

**Fig. 7 *Lipe* is expressed in the retina and RPE of C57BL/6J (B6J) mice. a** Western blot analysis of retina and RPE-choroid from *Lipe*$^{+/+}$ and *Lipe*$^{-/-}$ mice included the following samples: 1. Testes (positive control, 40 μg), 2. *Lipe*$^{+/+}$ retina #1 (30 μg), 3. *Lipe*$^{+/+}$ retina #2 (30 μg), 4. *Lipe*$^{+/+}$ RPE-choroid #1 (20 μg), 5. *Lipe*$^{+/+}$ RPE-choroid #2 (20 μg), 6. *Lipe*$^{-/-}$ retina (40 μg), 7. *Lipe*$^{-/-}$ RPE-choroid (40 μg), 8. *Lipe*$^{-/-}$ testes (40 μg). Antibodies for beta-actin and GAPDH were used as loading controls. **b** RNAscope in C57BL/6J (B6J) retina using a probe for *Lipe* demonstrated a strong signal in the retina, particularly in the outer nuclear layer. There was also some signal in the inner nuclear layer and RPE. **c** A B6J testis was used as a positive control for the *Lipe* probe. **d** An *Sfxn3* probe in B6J retina[21] shows a very different staining pattern compared to the *Lipe* probe; the staining was strongest in the inner nuclear layers for the *Sfxn3* probe. **e** A DapB-bacterial dihydrodipicolinate reductase probe was used as a negative control in B6J testis. Scale bars for **b** and **d** = 50 μm, Scale bars for **c** and **e** = 75 μm.

low ($p = 8 \times 10^{-6}$) and the moderate intensity ($p = 1.3 \times 10^{-4}$) stimuli (Fig. 9b). However, there was no difference in the b-wave at either intensity at this age (Fig. 9c). ERG testing was repeated in 1-year-old mice, which corroborated a significant decrease in a-wave (Fig. 9d). Also, at this age we found a statistically significant decrease in the b-wave when applying the higher light stimulus (Fig. 9e). Since the ERG a-wave is a negative deflection resulting from hyperpolarization of the photoreceptors, these data suggest that *Lipe* deficiency leads to a functional defect in the photoreceptors. The ERG b-wave includes the depolarization of inner nuclear layer cells including bipolar cells and Müller cells. The fact that we only found a difference in the b-wave signal intensity in older mice and only when using the higher intensity stimulus suggests again that the main impact of *Lipe* deficiency is in the photoreceptor layer and not the inner nuclear layer.

In order to corroborate our findings regarding decreased retinal function, a separate measure of visual function was used: optokinetic responses. Using the OptoMotry system (Cerebral Mechanics, Inc., Lethbridge, AB, Canada) the spatial frequency threshold, a measure of visual function, was determined by a masked investigator. A significant decrease in threshold was identified in *Lipe*$^{-/-}$ compared to *Lipe*$^{+/+}$ mice (Fig. 9f, $p = 0.006$).

As described above, while we observed some changes in the RPE of *Lipe*$^{-/-}$ mice, these were mild and mostly due to increased phagocytosis of photoreceptor debris. Thus, we were not expecting significant differences in c-waves on ERG. Indeed, we did not find any difference in the scotopic c-wave ERG responses in *Lipe*$^{-/-}$ mice vs WT (Supplementary Fig. S14a). We also tested photopic ERG responses to determine if there were any cone-specific functional abnormalities in *Lipe*$^{-/-}$ mice

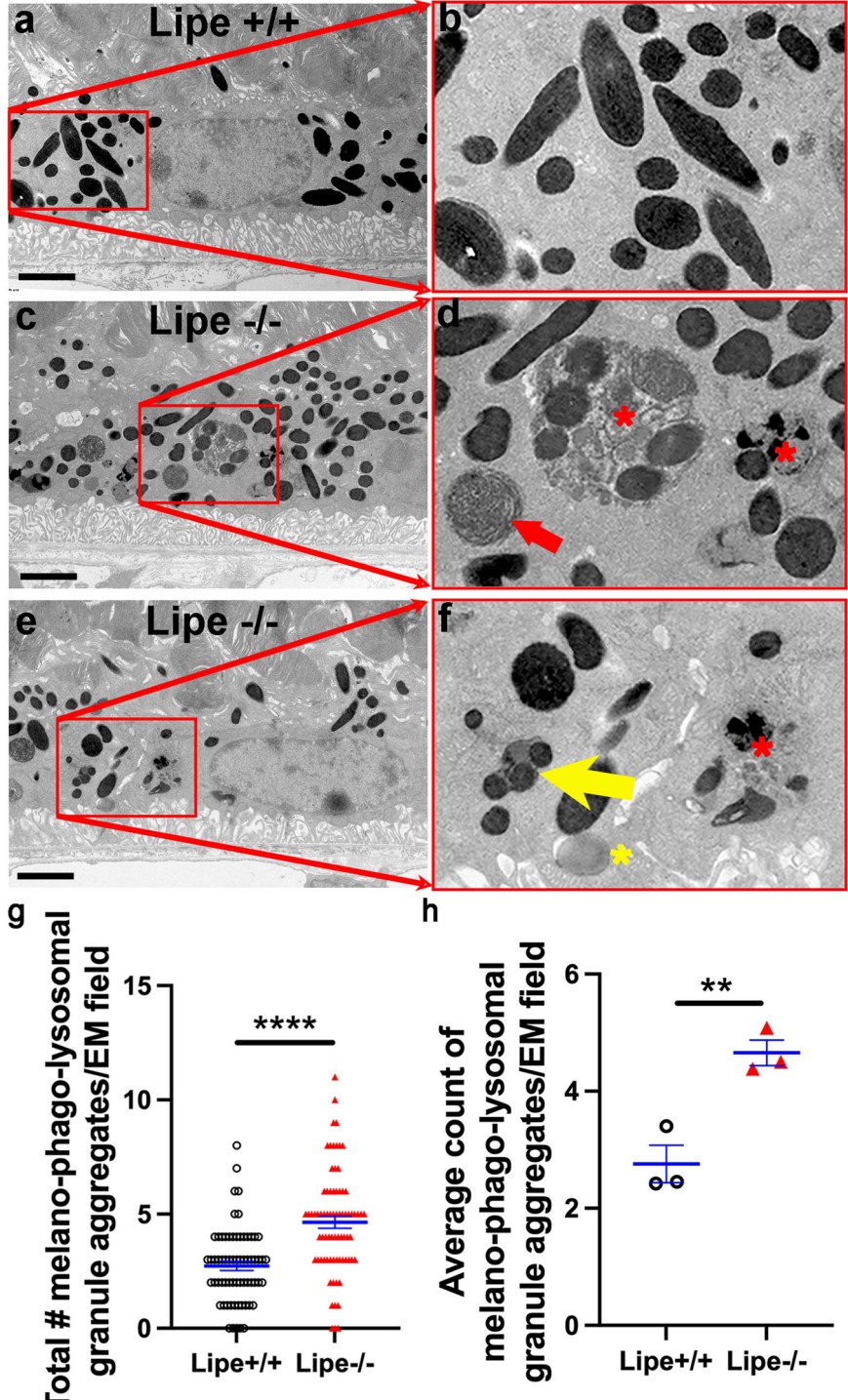

**Fig. 8 Intracellular RPE granule aggregates are increased in *Lipe*<sup>−/−</sup> eyes compared to *Lipe*<sup>+/+</sup> eyes.** Representative TEM images of the RPE of a *Lipe*<sup>+/+</sup> (**a**) and two *Lipe*<sup>−/−</sup> (**c, e**) eyes are shown (scale bars, 2 μm). Magnified views are shown in panels **b, d** and **f**, respectively. TEM images were systematically analyzed by a masked investigator who counted and added all phagolysosomes (red star), phagosomes (red arrow), melanofuscin (yellow arrow), and lysosomes (yellow star). The average number of melano-phago-lysosomal granule aggregates per TEM field (**g**), and per eye (**h**) were graphed for *Lipe*<sup>+/+</sup> (n = 66 TEM fields in **g** and 3 eyes in **h**) and *Lipe*<sup>−/−</sup> (n = 76 TEM fields in **g** and 3 eyes in **h**). In both of these analyses, a statistically significant increase in granule aggregates was seen in *Lipe*<sup>−/−</sup> mice compared to the control group. Data are presented as Mean ± SEM. Two-tailed Student's *t* test. **$p < 0.01$, ****$p < 0.0001$.

(Supplementary Fig. S14b, c). We were not able to find statistically significant differences in *Lipe*<sup>−/−</sup> mice, suggesting that most of the impact in photoreceptors is involving rods rather than cones. Moreover, while we had already noticed on OCT that the inner retina of *Lipe*<sup>−/−</sup> mice did not suffer any thinning, we also added oscillatory potential testing, which is known to correlate best with inner retinal function. The results showed similar signals in *Lipe*<sup>−/−</sup> and *Lipe*<sup>+/+</sup> mice (Supplementary Fig. S14d), confirming that the inner retina appears to be functioning well in *Lipe*<sup>−/−</sup> mice.

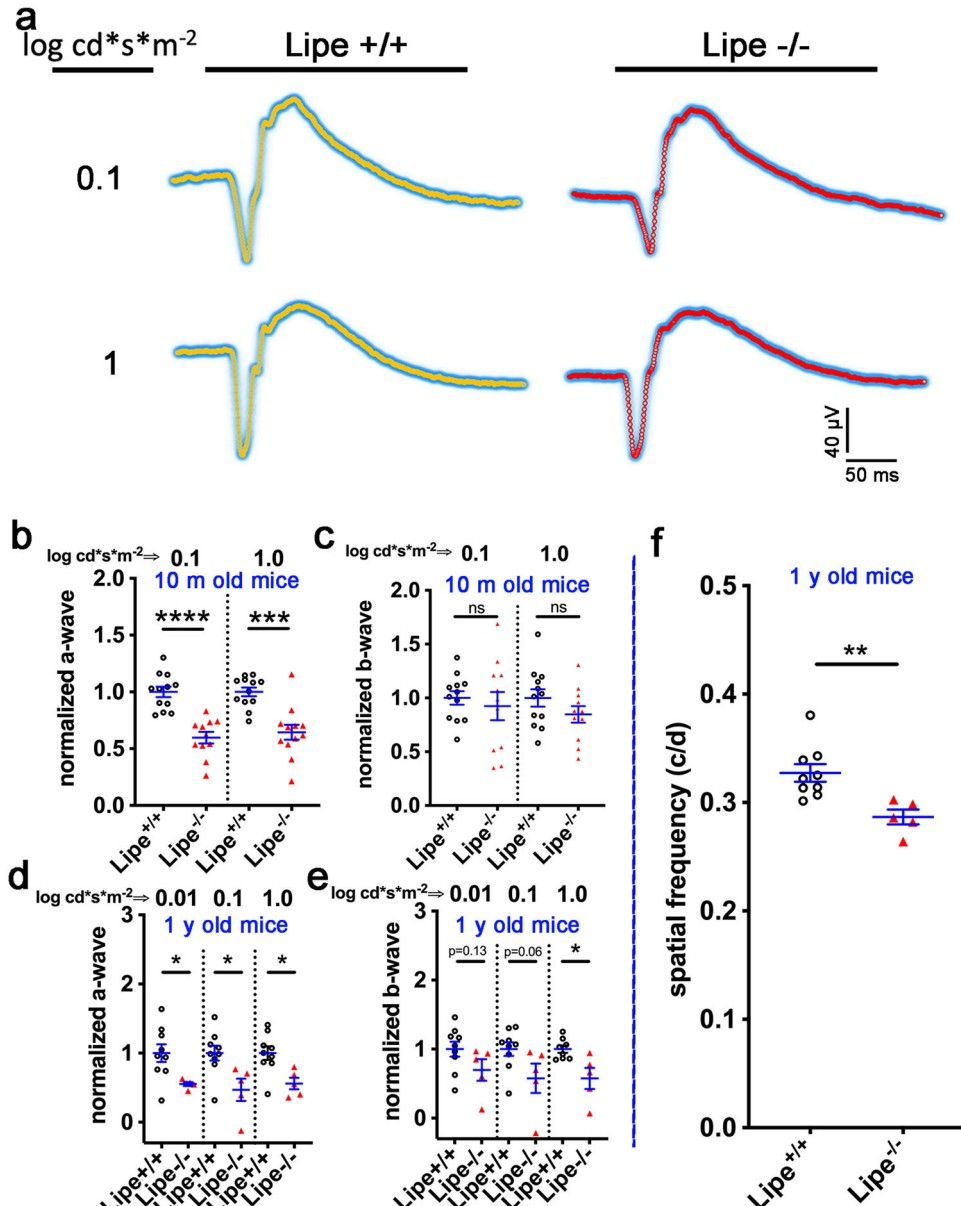

**Fig. 9 $Lipe^{-/-}$ mice develop functional deficits on both ERG and optokinetic responses.** Average ERG tracing from $Lipe^{+/+}$ ($n = 10$ mice) and $Lipe^{-/-}$ ($n = 12$) mice after two different light stimuli (**a**). Scotopic Ganzfeld ERG a-wave (**b**) and b-wave amplitudes (**c**) were measured at 0.1 and 1 log cd.s.m$^{-2}$ flash intensities in 10-month-old $Lipe^{+/+}$ and $Lipe^{-/-}$ mice. A significant reduction was observed in the ERG a-wave signals of $Lipe^{-/-}$ mice compared to $Lipe^{+/+}$. No significant difference was observed in the b-wave signals using these parameters. In contrast, ERG testing in 1 y/o mice ($n = 9$ $Lipe^{+/+}$ and 5 $Lipe^{-/-}$ mice) demonstrated decreased signals in both the a-wave (**d**) and the b-wave (**e**). Confirmation of the visual deficit in $Lipe^{-/-}$ mice was obtained using measurements of spatial frequency thresholds with the OptoMotry system (**f**). One-year-old $Lipe^{-/-}$ mice ($n = 5$) showed a significant decrease in optokinetic responses compared to age-matched $Lipe^{+/+}$ mice ($n = 9$). Data are presented as Mean ± SEM. Two-tailed Student's $t$ test. *$p < 0.05$, **$p < 0.01$, ***$p < 0.001$, ****$p < 0.0001$.

**$Lipe^{-/-}$ mice develop systemic lipid abnormalities and decreased weight.** Since our $Lipe^{-/-}$ mice have a global deficiency in this enzyme, and since other groups have shown that $Lipe$ deficiency can lead to some systemic lipid abnormalities, we decided to look at general systemic changes in our mice. We first looked at weight. The weight of each experimental mouse was recorded at multiple timepoints (Supplementary Fig. S15a). No statistically significant differences in weight were seen in either female or male mice up to 6–8 months of age. We did observe a trend ($p = 0.053$) towards a reduction in weight in male $Lipe^{-/-}$ mice compared to WT at 6–8 months. However, at 9–14 months, while female $Lipe^{-/-}$ mice continued to gain weight normally, $Lipe^{-/-}$ male mice

show a significant decrease in weight compared to controls (32.9 g vs 38.4 g, or a 14% decrease in weight, $p = 1.9 \times 10^{-5}$).

We then analyzed the levels of total cholesterol (TC), high density lipoprotein (HDL), and triacylglycerols (TAGs) in serum in wild type and $Lipe^{-/-}$ mice. As shown in Supplementary Fig. S15b, there was a statistically significant increase in total cholesterol for both male (39% increase, $p < 0.05$) and female (31% increase, $p < 0.05$) $Lipe^{-/-}$ mice compared to WT. Furthermore, male $Lipe^{-/-}$ mice (but not females) showed a significant increase in HDL compared with WT (42% increase, $p < 0.05$). Male $Lipe^{-/-}$ mice also had increased HDL compared to female KO mice (37% increase, $p < 0.05$)

Prior studies on $Lipe^{-/-}$ mice reached conflicting conclusions regarding weight, with one study finding no differences in weight in $Lipe^{-/-}$ mice up to 6 months of age[42], while another one reported a decrease in weight in $Lipe^{-/-}$ mice by 4 months of age, but did not compare female vs male mice[43]. Our study suggests that $Lipe$ deficiency mainly affects male mice and increasingly with age. Prior studies had shown an increase in total serum cholesterol in both male and female $Lipe^{-/-}$ mice[42], which we corroborate here. We also corroborate the previous finding of unchanged serum triacylglycerol levels in $Lipe^{-/-}$ mice[42]. Finally, our study suggests for the first time that serum HDL is elevated specifically in male $Lipe^{-/-}$ mice.

**Retinal and RPE lipid abnormalities identified in $Lipe^{-/-}$ mice.** We then sought to better understand the specific pathogenetic mechanisms involved in the retinal degeneration induced by the absence of HSL. Given the fact that this is a key enzyme in several lipid catabolism pathways that may be active in the retina, including those involving triacylglycerols, cholesteryl esters and retinyl esters, we decided to study the lipidome of both retina and RPE/choroid in $Lipe^{-/-}$ vs $Lipe^{+/+}$ mice. After lipid extraction from retina vs RPE-choroid-sclera, we conducted lipidomic analyses of the tissue specimens using UPLC-MS as described in the Materials and Methods section. High resolution mass spectra of the triacylglycerol fractions are shown for each of the samples (Fig. 10a). The retina samples of $Lipe^{-/-}$ showed an increased abundance of long-chain triacylglycerols (purple oval in Fig. 10a). Extracted ion chromatograms of triacylglycerols, diacylglycerols, cholesterol and cholesteryl esters are shown in Fig. 10b. Furthermore, a striking difference in the lipid composition between $Lipe^{-/-}$ and $Lipe^{+/+}$ mice was seen in an untargeted analysis of both retina and RPE tissues, as documented by OPLS-DA (Orthogonal Projections to Latent Structures Discriminant Analysis). All $Lipe^{-/-}$ retinas can be seen clustered far from the $Lipe^{+/+}$ retinas (Fig. 10c), and the same is seen for RPE-choroid-sclera tissues of KO vs WT mice (Fig. 10d). Interestingly, the one possible $Lipe^{+/+}$ retina outlier was an eye in which the vitreous gel was not completely separated from the retina during dissection.

Further analysis focused on two out of the three major lipid groups affected by $Lipe$: 1) triacylglycerols and diacylglycerols, and 2) free cholesterol and cholesteryl esters. When comparing triacylglycerols to diacylglycerols in $Lipe^{-/-}$ vs. $Lipe^{+/+}$ retinas, we found that triacylglycerols were increased relative to diacylglycerols in $Lipe^{-/-}$ retinas. This was the case whether we plotted the ratio of TAGs to total TAGs + DAGs (Fig. 10e, $p = 0.008$) or if we plotted the ratio of TAGs to DAGs (Fig. 10f, $p = 0.012$). The same comparisons in RPE-choroid-sclera samples did not show any significant differences ($p = 0.573$ for TAG/(TAG + DAG) and 0.269 for TAG/DAG). When looking at cholesteryl esters, we found that the ratio of cholesteryl esters to total cholesterol [CE/(Chol+CE)] was significantly increased in $Lipe^{-/-}$ retinas compared to WT retina ($p = 0.034$). Meanwhile, this parameter was again not different when comparing the RPE-choroid-sclera of $Lipe^{-/-}$ vs WT mice ($p = 0.737$).

Taken together these findings indicate that the absence of HSL causes noticeable lipid catabolism abnormalities in both the retina and RPE. Furthermore, it seems that the role of HSL in the hydrolysis of TAGs and cholesteryl esters may be more important (or irreplaceable) in the retina than in the RPE-choroid-sclera.

## Discussion
Data from patients with AMD, retinal dystrophies, and diabetic retinopathy indicate an important role of immune cells, including microglia, in the pathogenesis of these retinal diseases[1]. The accumulation of drusen components provides an environment rich in chemoattractants for microglia and leads to their translocation to the subretinal space in AMD[2,4]. The involvement of microglia in the activation of the $NLRP3$ inflammasome and the promotion of proinflammatory cytokine secretion has been confirmed in in vitro and animal studies[11,12,14]. In patients with retinal dystrophies like retinitis pigmentosa, it has been shown that microglia become activated in response to signals from degenerating rod photoreceptors and migrate to the outer retinal layers[4]. There, they participate in the phagocytosis of debris and dying cells and secrete proinflammatory factors. Mouse models of retinal degeneration (e.g. rd1, rd7, rd8, and rd10 models) confirm many of these conclusions[9,10,13,15], but make it clear that the role of microglia may also be homeostatic, depending on both stimuli and anatomical location within the retina[7,20]. Activated microglia are observed at all the stages of human diabetic retinopathy[3,8] and also feature prominently in many animal models of the disease[44,45]. Finally, accumulations of activated microglia are also seen in a variety of animal models of retinal degeneration, including light-induced retinal degeneration and models based on complement dysregulation[34,46,47].

The pathways regulating immune surveillance, cell trafficking, and neuroinflammation in the retina are not well understood. A large number of molecules and processes have been implicated, ranging from chemokines involved in chemotaxis, cytokines involved in activation, factors that regulate oxidative stress and complement activation, and immunoregulatory proteins. In such a complex biological system, the unbiased nature of a forward genetics approach is particularly valuable in identifying genes affecting these immune cell processes. Furthermore, the accumulation of subretinal microglia, visible as or correlated with the accumulation of fundus spots, can serve as a marker for retinal pathology and thus as a screen for genes essential to retinal homeostasis. Our approach here has two important advantages relative to all prior forward genetics studies of the retina: 1. We are systematically applying a semiquantitative fundus spot scale to fundus photographs, and 2. Our pipeline is the only one in which all mice screened are G3 mice that have been pregenotyped at all mutant loci. Our unbiased identification of 6 gene-phenotype associations to retinal pathology with strong literature support using our fundus spot scale screen is proof of concept supporting the efficacy of our approach. We identified other associations that had not been reported in the literature at the time of the screening. From those, we first concentrated our efforts on the gene $Lipe$, partly because the fundus spot scale was the only parameter leading to its identification.

In order to confirm our findings in ENU-mutagenized mice and also to explore the role of $Lipe$ in retinal homeostasis, a CRISPR-generated $Lipe^{-/-}$ mouse line was generated. Imaging of the retinas on these mice confirmed an early and prominent accumulation of fundus spots. Furthermore, we found a similar widespread accumulation of hyperautofluorescent spots in these mice. We were also able to show that $Lipe^{-/-}$ mice have increased accumulation of subretinal Iba1+/CD16+/TMEM119+/CCR2− cells consistent with activated microglia. It can be argued that microglia migrating to the subretinal space are by definition showing some level of activation[48–50]. But our findings of well-accepted morphological signs of activation and co-staining with CD16, a marker of microglial activation[10,34,51,52], further support this conclusion. Finally, Spectralis images seemed to indicate that the yellow spots seen on fundus photographs correlated with small hyperreflective spots right above the RPE/IZ band on OCT. A one-to-one correlation between fundus spots and Iba1+ subretinal microglia is not possible for several reasons, including the fact that fundus photos are 2-dimensional representations of a spherical surface, while the flat mount utilizes radial cuts to

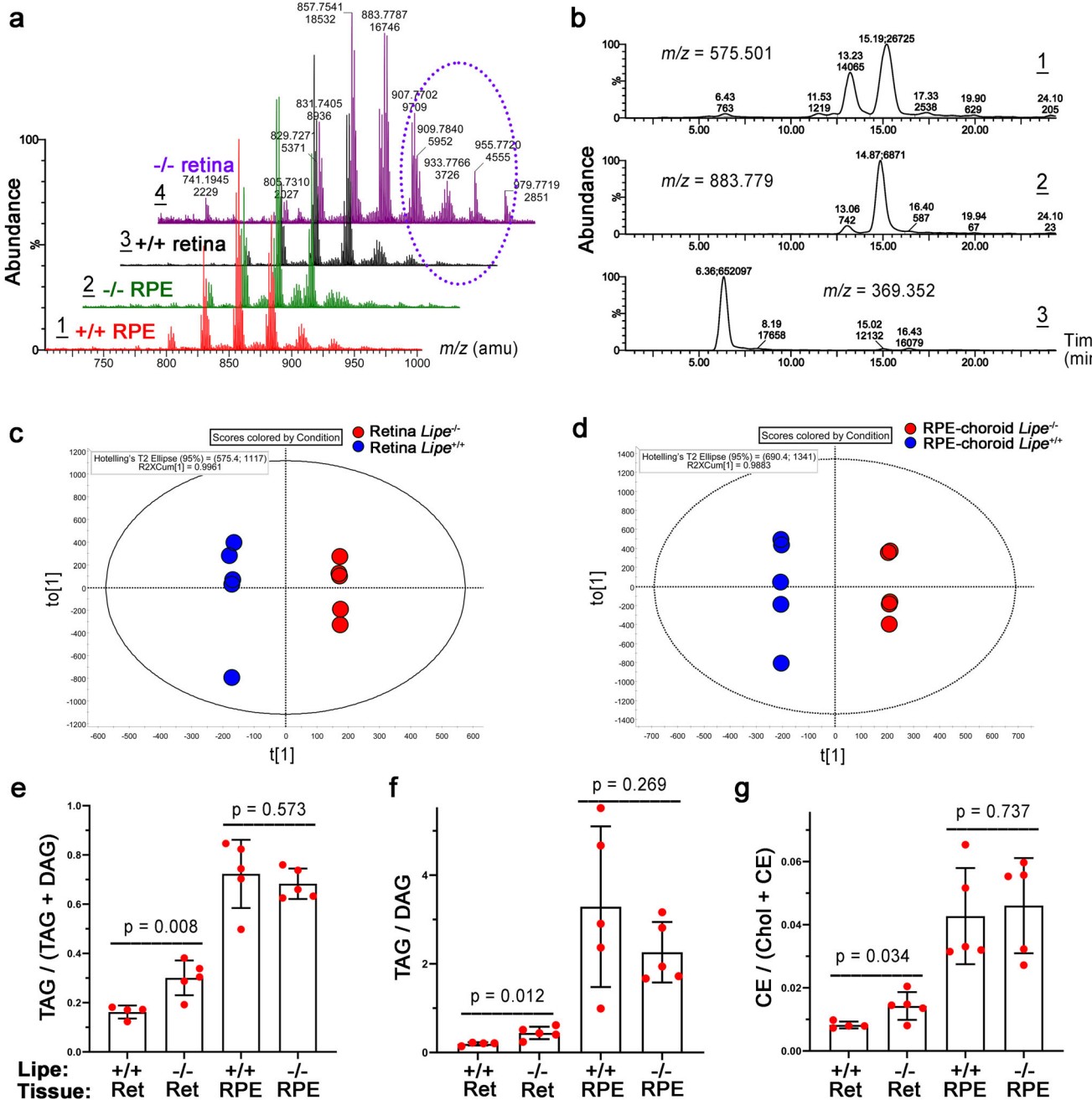

**Fig. 10 Lipidomic analyses of *Lipe*$^{+/+}$ and *Lipe*$^{-/-}$ retina and RPE/choroid mouse tissues (*n* = 5 per group). a** Mass spectra of the TAGs of *Lipe*$^{+/+}$ RPE/choroid (spectrum <u>1</u>), *Lipe*$^{-/-}$ RPE/choroid (<u>2</u>), *Lipe*$^{+/+}$ retina (<u>3</u>) and *Lipe*$^{-/-}$ retina (<u>4</u>) showing different peaks in *Lipe*$^{-/-}$ retina (purple oval). **b** Trace 1: An extracted ion chromatogram (EIC) of an analytical ion with *m/z* 575.501 that can originate from either DAGs or TAGs. The existence of several chromatographic peaks with different retention times (RTs) implies the presence of several parent compounds. The peak with RT 6.43 min is a true DAG with an elemental composition $C_{37}H_{66}O_4$ detected as an $(M - H_2O + H)^+$ ion, while peaks with RT 13.23 min and 15.19 min are spontaneously in-source fragmented TAGs with the same $C_{37}H_{66}O_4$ moiety. Trace 2: An EIC of an ion *m/z* 883.779. The main peaks with RT 13.06 min and 14.87 min belong to a TAG $C_{57}H_{102}O_6$ detected as an $(M + H)^+$ adduct. The existence of two major chromatographic peaks implies the presence of two major isobaric structural isomers of the compound. Trace 3: An EIC of an analytical ion *m/z* 369.352. The ion is characteristic of both Chl and CEs. The main peak with RT 6.36 min belongs to Chl ($C_{27}H_{44}O$) detected as an $(M - H_2O + H)^+$ adduct. Note the presence of smaller peaks at 15.02 min and 16.43 min, which reveal the presence of, at least, two CEs. All LC—MS peaks are labeled as (RT; Peak area). **c** The *Lipe*$^{+/+}$ and *Lipe*$^{-/-}$ retina samples were compared using OPLS-DA. **d** The *Lipe*$^{+/+}$ and *Lipe*$^{-/-}$ RPE/choroid samples were compared and demonstrated a similar grouping pattern. Inactivation of *Lipe* caused noticeable, statistically significant changes in the balance between TAGs and DAGs (Panels **e** and **f**), and Chl and CEs (Panel **g**) in the retina, but not in the RPE/choroid. Calculated ratios in panels **e**-**g** were obtained for *Lipe*$^{+/+}$ and *Lipe*$^{-/-}$ (*n* = 5 for each group). Data are presented as Mean ± SEM. Two-tailed Student's *t* test.

flatten the spherical retina in a nonuniform manner. Still considering all these data together, it is tempting to speculate that the yellow fundus spots in the $Lipe^{-/-}$ mice are responsible for the hyperreflective subretinal spots on OCT, and that they represent activated subretinal microglia. Of interest, the size of the fundus spots in fundus photos, subretinal hyper-reflective spots on OCT and Iba1+ subretinal microglia appear to be similar. The fundus spots vary in size but are in average 20–30 µm in diameter. In autofluorescence, the spots are smaller (around 15–20 µm). However, it should be noted that if the autofluorescence is due to intracellular material, the size of the autofluorescent signal would be expected to be smaller in diameter than the entire cell. The subretinal hyperreflective OCT spots that we show in Supplementary Fig. S6 are around 40 µm in size, but we have chosen some of the larger spots, so this is within the range of the larger fundus spots (which can be up to 40–50 µm in size). While the entire size of subretinal microglia can be up to 40–50 µm, the cell bodies of the activated subretinal microglia is around 20–25 µm. All of these sizes are within range considering the fact that each of these modalities may highlight different properties of the cell (color in fundus photos, may not exactly correspond to autofluorescence, or to hyperreflectivity).

Despite the fact that our OCT screens did not pick up an association of $Lipe$ to retinal pathology, we decided to evaluate this further in the CRISPR-generated $Lipe^{-/-}$ mouse line. We found definite evidence of a progressive retinal degeneration affecting the outer retina, as documented by our analyses of OCT images, histology and ERG. The relatively moderate nature of the degeneration in combination with the fact that we only had two mice homozygous for the reservoir allele in our forward genetics screen explain the lack of a statistically significant hit using the OCT parameters. This, however, highlights an interesting fact. In prior work we identified a gene ($Sfxn3$) that when mutated leads to very prominent outer retinal thinning, yet there is no notable fundus spot accumulation. In contrast, in the $Lipe^{-/-}$ mouse there is prominent accumulation of fundus spots and of activated subretinal microglia, but only mild/moderate retinal thinning. This raises at least three potential scenarios: 1. the microglia are becoming activated by debris generated by the retinal degeneration, 2. the HSL deficiency is causing a metabolic defect that is directly triggering the microglial activation, which in turn is causing damage to the outer retina, or 3. the HSL-induced metabolic abnormality is causing both the retinal degeneration and the microglial activation. The RNAscope findings indicating that within the retina $Lipe$ is most prominently expressed in the ONL, combined with the fact that structurally (OCT and H&E) the retinal degeneration is mostly affecting the outer retina would suggest that the lipid metabolism defects may be causing damage to the photoreceptors. This conclusion is also supported by the fact that the strongest ERG abnormality seen in $Lipe^{-/-}$ mice is a decrease in a-wave amplitude. While it is tempting to speculate then that the activation of the microglia is the result of the debris generated by the photoreceptor degeneration, this would not explain why the more severe photoreceptor degeneration in $Sfxn3^{-/-}$ mice does not lead to notable microglial activation. Thus, we would hypothesize that the lipid abnormalities caused by the HSL deficiency may also be contributing to microglial activation.

The TEM findings in $Lipe^{-/-}$ mice likely indicate an abnormality in the lysosomal degradation process in RPE cells. However, this may be due to: 1. deficiencies on the RPE organelles or enzymes leading to blocked degradation pathways and thus accumulation of granules, or 2. Increased photoreceptor debris overwhelming the degradation pathways. Thus, the TEM findings may be due to primary pathology in the RPE or secondary to pathology in the outer retina.

Although the expression and effects of the HSL enzyme have been studied in different tissues previously, mostly testes and adipose tissues[42,53,54], the role of this enzyme in the retina is still unclear. Of interest, a recent publication suggests that HSL may be expressed in neuronal synapses in the brain. HSL has several isoforms, but the most common ones are a long form of around 130 kDa (testes, pancreatic β cells, and to a lesser extent adipocytes) and a short form of 80–90 kDa (adipocytes, testes, brain cortex, myocytes, and hepatocytes)[55,56]. These different isoforms are generated by the use of alternative translational start codons[56]. HSL is an intracellular neutral metabolic serine hydrolase[57] that is thought to play an important role in three metabolic pathways: 1. conversion of cholesteryl esters to free cholesterol, 2. release of free fatty acids from TAGs, and 3. hydrolysis of retinyl esters to generate retinol. Of interest, Pikuleva's lab[58] has shown that cholesteryl ester hydrolysis occurs in the retina and that 3 enzymes ($Nech1$ in the endoplasmic reticulum and microsomes, $Lipa$ in lysosomes and $Lipe$ in the cytosol) may be important in this process. It was not clear which form of $Lipe$ was active in their studies[58]. Regarding the release of fatty acids from TAGs, while other enzymes are more important for the release of the first fatty acid (e.g. adipose triglyceride lipase), or the third fatty acid (e.g. monoacylglycerol lipase), HSL seems to be the most important for the conversion of diacylglycerols to monoacylglycerols. Triacylglycerol hydrolysis occurs in the RPE, but there is less data on its importance in the retina, and no published data on the role of $Lipe$ in TAG hydrolysis in either the retina or RPE. Finally, it seems that in hepatocytes and adipose tissue HSL is also important in the hydrolysis of retinyl esters. While retinyl ester hydrolysis is important for the visual cycle, enzymes involved in this process in the liver and retina include patatin-like phospholipase domain-containing 3 (PNPLA3)[59] and neutral cholesterol ester hydrolase 1[60]. It is again unclear the relative importance of $Lipe$ in retinyl ester hydrolysis in the retina and RPE.

With the goal of better understanding the mechanisms of disease underpinning the retinal degeneration in $Lipe^{-/-}$ mice, we have started analyzing the effects of this mutation on the lipid profile of the retina and RPE-choroid-sclera. Importantly, a striking clustering of $Lipe^{-/-}$ vs $Lipe^{+/+}$ samples in OPLS-DA analysis clearly shows that the lipidomes of both the retina and the RPE-choroid-sclera are markedly altered by the absence of HSL. In other words, HSL has an essential, non-redundant, function in both retina and RPE/choroid lipid homeostasis. We have first looked at the effect of $Lipe$ deficiency on cholesteryl ester and TAG hydrolysis and found that there is a relative increase in the precursors of both of these pathways (cholesteryl esters and TAGs/DAGs) in $Lipe^{-/-}$ mice, but only in retinal tissues. We are now preparing to study the impact of $Lipe$ ablation on retinyl esters. Since the differences on OPLS-DA analysis of the lipidome of RPE-choroid-sclera samples of $Lipe^{-/-}$ vs. $Lipe^{+/+}$ mice mimicked those seen on retinal tissues, and since the RPE-choroid-sclera did not show differences implicating defects in the cholesteryl ester or TAG hydrolysis pathways, we speculate that we will find important differences in the retinyl ester composition in $Lipe^{-/-}$ vs $Lipe^{+/+}$ retinas and RPE-choroid-sclera. Also, since HSL is involved in the hydrolysis of all three fatty acids in TAGs, we would expect that absence of HSL would lead to a relative accumulation of TAG > DAG > MAG. We did observe the TAG > DAG accumulation on $Lipe^{-/-}$ retinas. However, the effect of HSL is strongest and least redundant in the hydrolysis of the second fatty acid. Thus, we are in the process of finalizing a protocol for experiments to measure MAGs in order to look for changes in the DAG/MAG ratios, the results of which will be reported separately.

While we were characterizing our $Lipe^{-/-}$ mice, Sollier et al.[61] described 3 patients that were examined due to lipodystrophy.

These patients were found to have biallelic null *LIPE* variants. In addition to their systemic manifestations, all three of these patients were found to have retinal abnormalities. One patient had subretinal autofluorescent deposits in the posterior pole at the age of 45, the second one had drusen-like deposits at the age of 67 and the third one had disseminated autofluorescent sub-retinal deposits at the age of 50. The OCT revealed a small focal pigment epithelial detachment and areas of paracentral outer retinal atrophy. These patients did not have visual complaints. Of note, although they did test for 10 common genes associated to retinal dystrophies, they did not do whole genome sequencing. Thus, the retinal findings could potentially be due to other genes or combinations of genes. However, the findings of prominent yellow fundus spots in our *Lipe*$^{-/-}$ mice are strikingly similar to the description given of the fundi in all 3 of these patients, which were described as having "countless sub-retinal yellowish deposits disseminated in the macula and the peripheral retina". The buildup of many hyperautofluorescent spots in the retina of *Lipe*$^{-/-}$ mice is another similarity. These findings provide strong support for the hypothesis that HSL deficiency is responsible for the abnormalities seen in both mice and humans. It is tempting to speculate that with age, a progressive outer retinal degeneration may become symptomatic in patients with *LIPE* null mutations. Moreover, it is possible that combinations of these or other *LIPE* allele variants with other genes predisposing to retinal dystrophies or degenerations (including macular degeneration), may exacerbate retinal pathology in some individuals. Our work shows that while *Lipe* mutations do not result in a devastating retinal dystrophy, HSL does play an essential role in retinal homeostasis and its absence leads to thinning of the outer nuclear layer and loss of visual function (both by ERG and optokinetic responses). While some parameters only decreased by about 10% (e.g. total retinal thickness), other parameters decreased by close to 20% (e.g. outer nuclear layer thickness on both OCT and # of nuclear layers) and even 30% (ERG a-wave). These findings indicate that the bulk of the damage affected the photoreceptors, and it was impactful. Moreover, quality of vision is much more complex than what tests like visual acuity and electrophysiology can account for. Finally, the retina is a very complex tissue that requires an exquisite orchestration of physiologic processes for proper function. In many multifactorial diseases an alteration of a combination of processes modulates disease severity. Since as we demonstrate, *Lipe* plays an important and irreplaceable homeostatic role in the retina, it will be of interest to study if *Lipe*-related processes are in some way involved in common multifactorial diseases like macular degeneration or diabetic retinopathy in which microglial activation and lipid dyshomeostasis are thought to play a role.

This work demonstrates that using a fundus spot scale as a screening tool on a powerful forward genetics pipeline can identify important gene-phenotype associations to retinal pathology that are not only relevant to mice but also to humans. It also specifically corroborates that *Lipe* and its protein product HSL are critical to retinal homeostasis. This is in itself interesting partly because it is known that the retina and RPE have multiple lipases, which one may expect could fully substitute for the *Lipe* deficiency. In fact, RPE lysosomes alone contain up to 40 hydrolytic enzymes[62]. Yet, we show that *Lipe* deficiency leads to dysregulation of lipid metabolism with clear lipidome differences. We speculate that these abnormalities lead to damage to the photoreceptors and also to activation of retinal microglia. However, further studies are needed to determine how the microglia are activated and if they have a role in the abnormalities seen. Using our forward genetics approach we have now identified several genes essential to retinal homeostasis, but the phenotype observed in their absence fall in very different parts of the spectra

of fundus spot accumulation and outer retinal thinning. This provides us with a unique tool to study the mechanisms of retinal microglial activation. Among other things, we plan to pursue studies comparing single cell expression profiles in the retinas of several of these mutants at time points flanking the detection of subretinal microglia looking for chemoattractant and other modulators of immune cell activity. Future studies will also involve depleting microglia in *Lipe*$^{-/-}$ mice and some of the other models we are generating in order to determine the role of subretinal microglia in the different pathologic processes we are observing. Specifically, we want to determine if microglia are helping return retina to homeostasis by clearing excess photoreceptor debris or causing damage by increasing inflammatory mediators.

## Methods

**Animals.** The University of Texas Southwestern Medical Center's (UTSWMC) Institutional Animal Care and Use Committee (IACUC) reviewed and approved all experiments and procedures used in this study. We followed the NIH's guide for the care and use of Laboratory Animals, the Association for Research in Vision and Ophthalmology's Statement for the Use of Animals in Ophthalmic Research, and other applicable international and national guidelines. For the forward genetics studies, germ line mutations were randomly induced using N-ethyl-N-nitrosourea (ENU) treatment of C57BL/6J (B6J) male mice (see below). *Lipe*$^{-/-}$ mice were generated using CRISPR/Cas9 mediated gene targeting on a wild-type B6J background as detailed below.

**ENU mutagenesis, whole exome sequencing and determination of candidate genes.** Our unbiased forward genetics protocol (summarized in a simplified diagram, Fig. 1) involves the generation of mutations randomly throughout the genome of male B6J mice using N-ethyl-N-nitrosourea (ENU)[21,22]. Germline mutations are identified in G1 male founders using whole exome sequencing. G1 male mice are bred to generate G3 mice, which are screened for phenotypes (see Fig. 1 of Wang et al.[22]). Each G3 mouse carries around 60 mutations. The zygosity of each mutation in G2 dams and in all G3 mice of each pedigree is determined before phenotypic screening by sequencing across pedigree-specific coding/splice site mutations using Ion Torrent AmpliSeq custom primer panels. In this work, the main phenotypic screening was fundus photography followed by scoring of yellow fundus spots using a special fundus spot scale (see below).

Automated meiotic mapping was performed by Linkage Analyzer software to detect statistically significant mutation-phenotype associations[22]. In brief, Linkage Analyzer, an R-based analysis program, performs automated computations of single-locus linkage for every mutation in the pedigree. The magnitude of a quantitative phenotype is correlated with genotype (REF, homozygous for reference allele; HET, heterozygous for reference allele and variant allele; or VAR, homozygous for variant allele) at each mutation site in all mice in the pedigree. In the setting of ENU-generated pedigrees we refer to mice as REF, HET and VAR to emphasize that these mice have many other mutations and that we are only referring to the zygosity of the specific allele in question. The software uses recessive, semi-dominant, and dominant linear regression models to assess linkage. The output of this automated mapping is a Manhattan plot of genotype-phenotype associations for every mutation in the pedigree. In the Manhattan plot, $-\log_{10}$ P values (y-axis) for an association to a given phenotype are plotted vs. the chromosomal positions of the mutations (x-axis) that had been identified in the G1 founders of each pedigree.

Genes were considered of interest if their Manhattan plot peaks met the following criteria: 1) the peak is above a horizontal line representing a threshold of $P = 0.05$ with Bonferroni correction, and 2) the peak is at least 3 logs higher than the second highest peak in the pedigree.

Candidate Explorer, a machine learning algorithm, was then used to obtain an estimate of the likelihood that the phenotype of interest really emanates from the gene in question[35].

**Fundus photography and fundus spot grading.** We anesthetized mice using Ketamine/Xylazine cocktail (100 mg/kg-5 mg/kg) before performing the fundus screening. We used a mixture (1:1, v/v) of Phenylephrine Hydrochloride solution 2.5% and Tropicamide Ophthalmic Solution 1% to dilate the pupils. Before imaging, we applied GenTeal liquid gel (Novartis, East Hanover, NJ, USA) to the surface of both eyes to prevent corneal dehydration and to serve as coupling lens. Both eyes from each mouse were visualized and fundus photographs were captured using Micron IV retinal imaging microscope (Phoenix Micron, Inc. Bend, OR). Investigators masked to the genotypic data of the mice graded the photos using a modified version of a previously reported fundus spot scale[33]. Specifically, a fundus spot score for each eye is determined based on the amount of white/yellow fundus spots present as follows: no spots (score 0), 1 to 10 fundus spots (score 1), equivalent of one fundus-quadrant of spots (score 2), two to three fundus-

quadrants of spots (score 3), and all four fundus-quadrants with spots (score 4). We then add the scores of both eyes for a final score of 0 to 8 for each mouse (9 potential values). The resulting scores were loaded into the Mutagenetix platform. Linkage Analyzer correlated this information with the known genetic information on mutations for each mouse, and generated Manhattan plots in which $-\log 10$ P values (y axis) are plotted vs. the chromosomal positions of the mutations (x axis) identified in the G1 founders of each pedigree.

For statistical analysis of mutation-phenotype associations, we entered phenotype data as either a categorical variable or a continuous variable. Ordered categorical variables (ordinal traits) are analyzed using binomial calculations. Continuous variables are analyzed using a linear regression model to assess linkage. We found that due to the large number of values in our scale (9 values), a continuous variable analysis can be applied to our data and is more sensitive, allowing the detection of more associations.

**Generation of $Lipe^{-/-}$ mouse line.** CRISPR-Cas9 mutagenesis was used to generate a $Lipe$ knock out ($Lipe^{-/-}$) mouse line according to protocols described for other genes before[21,22]. Briefly, super-ovulated female C57BL/6 J mice were mated overnight with C57BL/6J male mice and fertilized eggs were collected the following day for in vitro transcription using $Lipe$ single guide RNA (sgRNA) and CRISPR-Cas9 technology. A total of 103 injected embryos were cultured in M16 medium (Sigma-Aldrich) at 37 °C in 5% $CO_2$. To generate mutant mice, 91 two-cell stage embryos were transferred into the ampulla of the oviduct (10–20 embryos per oviduct) of pseudo-pregnant Hsd:ICR (CD-1) female mice (Harlan Laboratories). Twelve pups were born, and 1 homozygous male was bred further to produce a knockout line containing the mutation shown in Supplementary Fig. S1. The mutation resulted in a 14 bp deletion (bp 220-233 of exon 2) of the $Lipe$ gene. This led to a frame-shift mutation beginning at amino acid 109 of the protein and terminating after the inclusion of 2 aberrant amino acids. Thus, the final peptide product was predicted to be 111 amino acids in length instead of the normal 802. While these mice were viable, male $Lipe^{-/-}$ mice have been reported to be infertile[63]. We generated a colony of $Lipe^{-/-}$ mice by breeding $Lipe^{-/-}$ female mice to $Lipe^{+/-}$ male mice. A colony of $Lipe^{+/+}$ mice was generated by first generating founder mice from crossing $Lipe^{+/-}$ males to C57BL/6J females. Thereafter $Lipe^{+/+}$ males were crossed to $Lipe^{+/+}$ females from these founders to maintain littermate controls.

**Preparation of retina and RPE-choroid-scleral flat mounts (RPE flat mounts), immunostaining and counting of subretinal microglia.** Mice (4 $Lipe^{-/-}$ and 4 control) were deeply anesthetized and both eyes were enucleated and fixed in 4% PFA at room temperature[10]. In brief, enucleated intact eyes were fixed in 4% PFA for 30 min, followed by an additional 30 min after removing the cornea. Then the retina and the RPE-choroid-sclera were separated and fixed for an additional 1 h. After washing 3 × 5 min in PBS, the RPE and retina flat mounts were double or triple stained at 4 °C overnight followed by incubation at RT for 2 h with the appropriate Alexa Fluor conjugated secondary antibodies. For microglia and infiltrating macrophage discrimination, we used a combination of anti-F4/80, anti-TMEM119, and anti-CCR2 antibodies. For detection of microglial activation, we used a combination of anti-Iba1, anti-CD16. (See Supplementary Table S2 for a list of antibodies, dilations used and other details).

For microglia counting the RPE flat mounts (4 $Lipe^{-/-}$ and 4 $Lipe^{+/+}$ mice) were observed under fluorescence microscopy at 20X magnification. Iba1+ cells were counted in selected fields at four quadrants (superior, inferior, nasal, and temporal) around the optic nerve head (ONH). Four 20X fields were selected in each of the central, paracentral, midperipheral, and peripheral regions. The Iba1+ cell counts from the four fields from each of the four regions of four flat mounts (see diagram in Fig. 3a) were used for subretinal microglia analysis and comparison between the two genotypes. For morphological analysis both the RPE and retina flat mounts (4 $Lipe^{-/-}$ and 4 $Lipe^{+/+}$ mice) were imaged using a Leica TCS SP8 confocal laser scanning microscope equipped with a Leica Application Suite X, LAS X, software (Leica Microsystems Inc.). Images were taken either at low (25X) magnification or at high (63X) magnification using a sequential scanning method.

**Image-guided OCT and measuring thickness of retinal layers on OCT images.** After anesthetizing mice and pupil dilation (see above), we obtained OCT images from both eyes using a Micron IV-OCT2 (Phoenix-Micron, Inc). Images obtained as described before[64] were used to determine retinal thickness measurements. The thickness of several retinal layers was measured using Fiji/ImageJ (https://doi.org/10.1038/nmeth.2019). Three measurements at 100 µm intervals were taken in the center of OCT image for each of the following four OCT parameters: 1) Total Retinal Thickness (TRT), 2) Ganglion Cell Complex (GCC), 3) Outer Nuclear Layer (ONL), and 4) Outer Retinal Thickness (ORT). Using the Straight-Line tool in Fiji/ImageJ, TRT was measured from top of Bruch's membrane (BM) to top of Internal Limiting Membrane (ILM). GCC was then measured from the top of the Inner Nuclear Layer (INL) to the top of the ILM. ORT was measured from the top of BM to the top of External Limiting Membrane (ELM). After sharpening the image to get better contrast, we then took our last measurement of ONL from the top of the ELM to the bottom of the Outer Plexiform Layer (OPL).

A separate instrument (Spectralis® OCT, Heidelberg Engineering, Heidelberg, Germany) was used for experiments trying to correlate fundus spots changes with time to OCT findings. Images were acquired according to the manufacturer protocols for registration and tracking changes in the fundus spots overtime.

**Outer nuclear layer measurements on H&E-stained retinal sections prepared by either a freeze substitution or a cryosection technique.** To corroborate our OCT retinal thickness data, we prepared histological sections of retinas from $Lipe^{-/-}$ and $Lipe^{+/+}$ mice ($n = 6$ per group) as described before[47]. Briefly, we collected right eyes of each mouse for freeze-substitution fixation and subsequent Hematoxylin and Eosin (H&E) staining. Sequential images of the H&E sections were taken at 20x magnification on either side of the ONH using a Leica DM2000 Upright Compound microscope (Leica Microsystems, Danaher Corporation Wetzlar, Germany,) equipped with an Optronics Microfire color CCD camera (Optronics, Goleta, CA, USA). The H&E images were opened in ImageJ and the ONL thickness was measured at 300 µm interval starting from the ONH on both directions. We took an average of three measurements within a 20 µm area at each interval. The number of cells in the ONL was also counted at the 300 µm interval by using the duplicate tool in image J which covered a rectangular area making sure we included three columns of nuclei consistently on one side of the 300 µm mark. The result was divided by 3 to report an average nuclear count per column.

For ONL measurements in cryosections, eyes were enucleated from 8-month-old mice ($Lipe^{+/+}$, $n = 4$ eyes, $Lipe^{-/-}$, $n = 4$ eyes). The eyes were placed in the same orientation using the optic nerve head as reference in an OCT compound and immediately frozen in liquid nitrogen. The frozen samples were then processed by a routine sectioning method. The cryosections were then used for H&E staining and ONL quantitation as described above.

**TUNEL assay and cone arrestin staining.** Freeze-substitution fixated eyes were paraffin processed without interceding hydration. They were embedded in sagittal orientation and sectioned at 5 µm by rotary paraffin microtomy according to established procedures. Serial paraffin sections were concomitantly prepared and checked by dark-field microscopy for mid-line ocular anatomy. The resulting sections were used for Terminal deoxynucleotidyltransferase-mediated UTP End Labeling (TUNEL) and for cone arrestin staining. Positive nuclei of retinal cells possessing nicked DNA were labeled with fluorescein according to methods of first report[65] and literature supplied with the DeadEnd Fluorometric TUNEL System (Promega Cat # G3250). Sections subjected to TUNEL were counterstained with propidium iodide. To check for the presence or absence of cone-photoreceptor clumping we stained retinal sections using anti-cone arrestin antibodies as a cone marker.

**Electroretinogram (ERG) analyses of the visual response in $Lipe^{-/-}$ and $Lipe^{+/+}$ mice.** A full-field scotopic ERG system (Celeris System, Diagnosys LLC, MA, USA) was used to record the responses of retina cells to light in $Lipe^{-/-}$ vs. $Lipe^{+/+}$ mice after dark adaption overnight for 16 h. ERG recordings were performed under a dim red light. After anesthesia and pupil dilation, each mouse was positioned on the ERG mouse platform, which is equipped with a thermal regulator. The full-field stimulators with built in electrodes to touch each eye were then placed. We recorded 2 channels with 10 sweeps to obtain 2 standard responses/parameters of scotopic ERG (a-wave, b-wave) for both eyes. The ERG was obtained in response to moderate (0.1 log cd.s.m$^{-2}$) and high (1 log cd.s.m$^{-2}$) flash intensities. The ERG analysis of visual response was obtained in response to low (0.1 log cd.s.m$^{-2}$) and moderate (1 log cd.s.m$^{-2}$) flash intensities. The inter-stimulus interval was 0.7 s and 60 s for low and high flash intensities, respectively. The flash duration was 1 msec. A similar protocol using 3 sweeps was used to measure c-waves. After 10 min of light adaptation at a setting of 3 log cd.s.m$^{-2}$, the photopic ERG measurements were obtained at 3 and 10 log cd.s.m$^{-2}$ flash intensities. Ten sweeps were recorded and averaged for each flash intensity. We analyzed the ERG data using Diagnosys Espion Software (Diagnosys, Inc) for comparison of retinal function between the $Lipe^{-/-}$ vs $Lipe^{+/+}$.

**Optokinetic testing of visual function.** Optokinetic testing was performed using the OptoMotry system (Cerebral Mechanics, Inc., Lethbridge, AB, Canada) in order to examine changes in visual acuity in $Lipe^{+/+}$ vs. $Lipe^{-/-}$ mice. Visual stimuli were displayed on four LCD screens placed around a central mouse stand[66,67]. Each mouse is tested individually while placed without restrain on the central stand. A visual stimulus consisting of a rotating vertical sine-wave grating is presented to the mouse and the optokinetic reflex is then recorded by manual tracking of head movements. To determine spatial frequency thresholds, an increasing staircase paradigm was utilized with 100% contrast. All recordings were done by a masked investigator.

**Immunoblotting and detection of $Lipe$ protein in different tissues.** We performed western blot to determine the expression of $Lipe$ protein in ocular tissues and testis (used as positive control) of $Lipe^{-/-}$ and $Lipe^{+/+}$ mice. Mice were euthanized (with ketamine overdose) and the eyes and testis were harvested for protein extraction[34]. Briefly, we dissected each eye on ice to separate cornea, retina, and RPE/choroid under a dissection microscope. After combining each tissue from

both eyes per mouse we homogenized the pooled tissues using a 1 ml Dounce tissue grinder in 200 µl cold T-PER tissue lysis buffer (Catalog No. 78510; ThermoFisher Scientific, Rockford, IL, USA) containing a protease inhibitor cocktail. The homogenized tissues were centrifuged at $12,000 \times g$ at 4 °C for 10 min. The supernatants were collected and the concentration of protein in each sample was determined using a BCA kit (Catalog No. 23225, ThermoFisher Scientific). Testes were dissected from $Lipe^{+/+}$ male mice to serve as positive control and homogenized similarly. For retina and RPE/choroid samples we used protein concentrators to obtain a desired protein amount (Pierce™ Protein Concentrator, 3 K MWCO, 0.5 ml). A known amount (20–40 ug) of each sample was prepared and boiled in SDS-BME for 5 min and loaded on a 4–20% Tris-Glycine gel (Cat. No. XP04205BOX, ThermoFisher Scientific). After electrophoresis for 80 min, the proteins were transferred onto a nitrocellulose membrane and blocked overnight in Intercept (PBS) blocking buffer (Cat. No. 927-70001, LI-COR Biosciences). After removing the blocking buffer and washing 3X in TBS-T, the membrane was incubated with HSL primary antibody (Cat. No. 4107S, Cell Signaling) diluted 1:1,000 in 5% BSA (in TBS with 0.02% $NaN_3$) overnight at 4 °C. After removing the primary antibody, the membrane was washed in TBS-T 3X and incubated with HRP chemiluminescent secondary antibody (Cat. No. 926-80011, LI-COR Biosciences) diluted 1:15,000 in 5% milk solution for 40 min at room temperature. Finally, after removing the secondary antibody solution and washing 3X, western HRP substrate (Cat. No. 34094, ThermoFisher Scientific) was added and the membrane was imaged on an Amersham Imager 600 (Amersham Biosciences, Piscataway, NJ).

**RNAScope in situ Hybridization (ISH).** Probes for RNAscope ISH were designed (Advanced Cell Diagnostics, Hayward, CA USA)[21] to detect the expression of Lipe RNA, Sfxn3 RNA (as a positive control/probe:Mn-Sfxn3-O1), and a negative control DapB probe (bacterial dihydrodipicolinate reductase mRNA) in $Lipe^{+/+}$ mouse retina or testes. After cardiac perfusion of 4-month-old mice with 4% PFA, eyes and testes were collected. Post-fixation was done overnight in the same buffer at 4 °C, followed by routine paraffin embedding. RNAScope ISH was then performed. The probes were run with Advanced Cell Diagnostics red chromogenic & fluorescent kits. Images were taken on a Leica DM2000 microscope (Leica Microsystems, Wetzlar, Germany) at 20X magnification using a Jenoptik Gryphax CCD camera, Texas Red filter, and acquired with Progress software (V.1.1.8.159).

**Transmission Electron Microscopy (TEM) imaging and analysis.** We collected eyes from deeply anesthetized $Lipe^{-/-}$ and $Lipe^{+/+}$ mice (see above) and the left eyes were processed for electron microscopy[21]. Eyes were fixed in 2% PFA and 2% glutaraldehyde in sodium cacodylate buffer followed by post fixation in 1% osmium tetraoxide. After trimming, dehydration and embedding in epoxy resin, 70-nm-thin sections were cut and stained with 2% aqueous uranyl acetate and lead citrate. The sections were then imaged with a JEOL 1200EX II transmission electron microscope (JEOL USA, Inc., Peabody, MA, USA)[21] with the help of UTSW Electron Microscopy Core.

For quantification of melanolipofuscin and auto/phagolysosome granules (MLaPL), we opened each micrograph in Fiji/ImageJ Software and counted all aggregates of fused organelles such as melanolipufuscin, auto/phagolysosomes in RPE cells of 3 $Lipe^{-/-}$ and 3 $Lipe^{+/+}$ mice (n = 20–26 TEM fields per group). We analyzed and reported the total numbers of these granules for each filed or averaged for each mouse. The thickness of the RPE, basal infoldings and Bruch's membrane were also measured.

**Measurement of axial length.** $Lipe^{+/+}$ (n = 21) and $Lipe^{-/-}$ (n = 18) mice were deeply anesthetized using a ketamine/xylazine cocktail. After enucleation, any residual adipose tissue around the optic nerve was removed under a dissecting microscope. Each eye was positioned on a glass plate to obtain the axial length. The length from the top of the cornea to the posterior sclera at the site of the optic nerve entry was measured three times using a digital caliper (Kynup Digital Caliper, eVatmaster Consulting GmbH, Germany). The measurements of both eyes of each mouse (6 total measurements per mouse) were averaged.

**Lipid extraction and analysis from retina and RPE/choroid samples.** Lipid standards such as cholesterol and cholesterol esters, triacylglycerols and diacylglycerols were purchased from MilliporeSigma (St. Louis, MO, USA), Avanti Polar Lipids (Birmingham, AL, USA) and Nu-Check Prep (Elysian, MN, USA) (see Supplementary Table S2). All other reagents used in the liquid chromatography experiments (see Supplementary Table S2) are available from MilliporeSigma, Burdick & Jackson (Muskegon, MI, USA), and ThermoFisher Scientific (Waltham, MA, USA).

Lipids were extracted from both retina and RPE-choroid-sclera samples (n = 5 for each) of $Lipe^{+/+}$ and $Lipe^{-/-}$ mice. In brief, mice were euthanized by cervical dislocation, eyes enucleated, and then placed in cold PBS. Cornea, lens, and vitreous[68] were surgically removed under a dissecting microscope. The retina was then separated from the RPE-choroid-sclera for each eye. The two retinas of each mouse were placed in a 2-mL glass HPLC sample vial with PTFE-lined cap filled with 0.5 mL of chloroform/methanol solvent mixture (2/1, vol/vol, CM2/1). The

samples were stored at −20 °C until the final extraction of lipids. RPE-choroid-sclera specimens (two from each mouse) were treated in a similar fashion. Retina and RPE/choroid lipids were extracted from the respective samples at room temperature thrice, each time with 1 ml of the CM2/1 solvent mixture for 15 min. The extracts were transferred stepwise into a glass crimper-style 300 µl HPLC vial, bringing the aliquots to dryness under a stream of purified nitrogen gas and keeping the vial warm at 36 °C. Finally, the combined oily residue was redissolved in 150 µl of iso-propanol in the same vial, crimped, and stored in a freezer until the analysis.

Extracted lipids were analyzed using an Acquity M-Class ultra-high performance liquid chromatograph (UPLC) and a Waters Synapt G2-Si high resolution quadrupole Time-of-Flight mass spectrometer (MS) equipped with an atmospheric pressure chemical ionization (APCI) IonSabre-II ion source with a Zspray/LockSpray housing (all from Waters Corporation; Milford, MA, USA).

The chromatograph was operated in the reverse phase UPLC mode (RP-UPLC). Briefly, each lipid sample was analyzed in two different experiments – initially on a $C_8$ (2.1 mm × 100 mm, 1.7 µm) RP-UPLC column, and then on a $C_{18}$ BEH (1.0 mm × 100 mm, 1.7 µm) Acquity RP-UPLC column (both from Waters, Corp.) using acetonitrile/iso-propanol solvent mixtures exactly as described earlier for other mouse ocular tissues[69]. The $C_8$ column was used in isocratic elution experiments, while the $C_{18}$ column was used in gradient elution experiments.

Lipid analytes (such as cholesterol, cholesteryl esters, tri-and di-acylglycerols) were detected in positive ion mode using the APCI technique[70–72]. Identification of selected major lipids was performed using their elemental composition derived from exact $m/z$ values ($+/−5$ mDa) in the EleComp routine of the MassLynx software (from Waters Corp./Nonlinear Dynamics; Milford, MA, USA), and UPLC retention times, which were compared with the retention times of authentic lipid standards.

The results of lipidomic experiments were analyzed using a Progenesis QI (from Waters/Nonlinear Dynamics; Milford, MA, USA), EZinfo (v.3.0.3.0; from Waters/Umetrics), and SigmaStat (v.3.5; from Systat Software, Inc., San Jose, CA, USA) software packages. For untargeted (i.e. unsupervised) analysis, the raw UPLC-MS data were processed in Progenesis QI using its Principal Component Analysis (PCA) feature. The data were normalized using total ion abundances determined for each run separately. Then, the data were imported into EZinfo for subsequent processing and analysis. The EZinfo's Orthogonal Projections to Latent Structures Discriminant Analysis (OPLS-DA) model implemented Pareto scaling. There were more than 200 unique variables (i.e. analytes with unique combinations of $m/z$ values and LC retention times, RTs) detected, exact identification of which goes beyond the scope of this manuscript and is to be reported separately.

**Serum lipid analysis.** Prior to euthanasia, 10–11-month-old $Lipe^{-/-}$ and $Lipe^{+/+}$ mice (n = 6, per group) were anesthetized, eyes were enucleated and blood was collected in 2 mL micro centrifuge tubes from the orbital sinus. Blood samples were allowed to clot at room temperature for 30 min. The samples were centrifuged at $1000 \times g$ for 10 min to separate the clot and serum was transferred into new microcentrifuge tubes for lipid analysis with the assistance of metabolic-phenotyping core of UT Southwestern Medical Center (http://touchstonelabs.org/metabolic-phenotyping-core/). Total cholesterol (TC), high-density lipoprotein cholesterol (HDL-C), and triacylglycerols (TAGs) were quantitated for comparisons of $Lipe^{-/-}$ versus age-and-gender matched $Lipe^{+/+}$ mice.

**Statistics and reproducibility.** Graphs and Statistical analysis were done using GraphPad Prism 9.4.1 and Microsoft Excel 16.63.1. Data are presented as the mean ± standard error of mean (SEM), except for Fig. 1 in which mean ± standard deviation is reported. For all figures, groups of measurements were taken from distinct samples, rather than from repeated measurements. Samples sizes are included in each figure legend. Normal distribution of the data was assumed. All comparisons were done between two groups using a two-tailed unpaired Student's $t$ test. A linear regression analysis was performed to check the trend of changes in the fundus spots and OCT parameters with age over time. A $p$ value $< 0.05$ was considered statistically significant with the null hypothesis that there are no differences between the two groups. For the analysis of Iba1+ cells in flat mounts we provide the $p$-value both including all values (black asterisks) and also excluding outliers (blue asterisks). We used the interquartile range (IQR) to determine outliers, which were defined as those outside of a range going from Q1–1.5*IQR to Q3 + 1.5*IQR, where Q1 and Q3 are the first and third quartiles, respectively.

**Reporting summary.** Further information on research design is available in the Nature Portfolio Reporting Summary linked to this article.

## Data availability

All data supporting this work are provided within the Article and Supplementary Files, or available from the corresponding authors upon request. Raw data for all figures is available at the Figshare repository[73], https://doi.org/10.6084/m9.figshare.c.6458461.v1. Genotyping sequencing data is also deposited in NCBI (Accession: PRJNA956881, ID: 956881). Unmodified blots are included as supplemental figures S16-S18.

## Material availability

*Lipe* KO mice are available upon request according to the UT Southwestern Medical Center MTA guidelines.

## Code availability

No custom code or algorithms were generated in this study. Candidate explorer is publicly accessible at https://mutagenetix.utsouthwestern.edu/linksplorer/candidate.cfm.

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

## Acknowledgements
The authors thank Emi Nakahara for her assistance with the Western blot assay and John Shelton at the Molecular Pathology Core for helping with histological specimens. Supported by NEI grant-1R01EY033181 (R.L.U-V.), NEI grant- R01 EY027349 (I.A.B.), National Eye Institute Visual Science Core Grant P30 EY030413, NIH grants R01 AI125581 (B.A.B.) and U19 AI100627 (B.A.B.), a VanSickle Family Foundation Grant (R.L.U-V.), a UTSW Pilot Synergy Grant (R.L.U-V.), and a David M. Crowley Foundation Grant (R.L.U-V.). Our research was also supported in part by the Josephine Long Biddle Chair in Age-Related Macular Degeneration Research, the Lillian and James Cain Endowment in Vision Loss, the Anne Marie and Thomas B. Walker Jr. Fund for Research on Macular Degeneration of the Retina, and the Department of Ophthalmology at UTSW.

## Author contributions
R.L.U-V. and B.A.B., conceived the project, designed experiments and analyzed data. S.Y., B.A., Y.Z., C.X.Z, L.G. and S.L. executed experiments and analyzed data. J.D.H. assisted with Western blot experimentation and analysis. M.T. and X.L. designed and generated the CRISPR transgenic mice. S.Y. and I.A.B. designed and performed lipidomic experiments and analyzed data. S.Y., B.A., E.M.Y.M., I.A.B., B.A.B. and R.L.U-V. wrote and edited the manuscript.

## Competing interests
The authors declare no competing interests.
