## [Peer Review File · Communications Biology]

Reviewers' comments:

Reviewer #1 (Remarks to the Author):

Title: Forward genetic screening using fundus spot scale identified essential role for Lipe in retinal homeostasis

Authors: Seher Yuksel,^{1¶} Bogale Aredo,^{1¶} Yeshumenesh Zegeye,¹ Cynthia X. Zhao,¹ Miao Tang and Xiaohong Li, John Hulleman,¹ Laurent Gautron,² Sara Ludwig,³ Eva M. Y. Moresco,³ Igor A. Butovich,^{1*} Bruce A. Beutler,^{3*} Rafael L. Ufret-Vincenty^{1*}

Summary: The authors present a great series of experiments that found several loci tightly linked to the appearance of fundus spots in mouse eyes. They have "...identified 14 genes that, when mutated, led to fundus spot accumulation." Here they characterized the most prominent locus and showed that the causative gene in the locus is Lipe, and its protein, hormone sensitive lipase E (HSL), that converts cholesterol esters to free cholesterol. The connection to cholesterol is intriguing.

There are a limited number of reports of lipodystrophy due to mutations in LIPE. There are apparently no reports of retinal or macular diseases in these patients.

Major points.

There needs to be a clearer and more thorough description of each of 14 mutations that cause appearance of excess fundus spots, in the body of the manuscript or in a table in the body of the manuscript.

Outside the obvious phenotype of many conspicuous spots on the fundus, other phenotypes (ONL thickness; ERGs, and so on) seem mild (though significantly decreased by statistical test, but still 90%+ of normal wildtype responses).

Effects on rods versus cones are not clearly distinguished.

Significance and meaning of findings:

1. Given these mild phenotypes even at relatively old age, what does this mean in terms of functional vision or disease etiology? The authors need to explain this better.

2. Could spots be a good thing? IE, If they are indeed activated microglia, could these spots be good microglia that promote repair and healing, or are they "bad" microglia that destroy cells that might be curable? The presumption is that these spots are bad and are promoting or causing degeneration of normal tissue, but maybe not... Explain how so (or not) in the manuscript.

Some additional experimentation might be warranted, and possible additional testing that might bring out more dramatic and clearer phenotypes that might be more informative:

- A. Is there a difference in vision at a spot versus anywhere else?
- B. More ERG analyses: C-waves, analysis of oscillatory potentials, Flicker ERGs, light adapted ERGs, Focal ERGs, and Pattern ERGs might be important or interesting to perform. Rods versus cones.
- C. Overloading the diet with specific lipids might provide a more distinctive functional phenotype.
- D. Other visual behavioral testing: OMRs. Visual acuity, contrast sensitivity. EOGs.
- E. Eye size: Are these mice myopic? Hyperopic? Could the spots and the slight thinning of the retina be functions of stretched retina?

What are the spots? While there seems to be evidence of co-registration between fundus spots and

activated microglia, is there a perfect one-to-one relationship? Are activated microglia the cause of the fundus spots? Maybe some of the spots are activated astrocytes, Muller cells, invading macrophages, defective RPE cells (or other defective cell types, for example, a dying cone photoreceptor) that are migrating?

Can the authors provide scale bars for the fundus images: How big are the spots compared to the spots by FAF, by SD-OCT, and by histology/confocal microscopy of Iba1 positive cells?

Do the same spots persist permanently, or do they change, move, or disappear and re-appear? Can the authors trace their movement over time?

Are the spots associated with a dead or dying cell, clump of dying cells? TUNEL staining might be informative.

Are these spots cellular or are they deposits of acellular debris or ECM? Oil slicks? Are they the beginnings of Drusen? Perhaps staining with some lipid stain (Oil red O, Nile red, etc.) be informative?

If the spots were "funny" cones, then staining sections with cone opsins might be informative.

Other questions: In Figure 6, H&E staining, outer segments and inner segments of the lipe KO mouse appear less organized and more ragged than in WT. Is this consistent? If so, please report this finding. Since OS's are so dependent on proper creation of membranes, and Lipe likely governs this it may be worth discussing.

Slight confounding factor: H&E's are generally done on paraffin embedded tissue (de-hydrated, re-hydrated). It appears that the authors fixed the eyes via a freeze-substitution technique (maybe probably -80 C Methanol: Acetic acid, 97:3; followed by ethanol.) The methanol, ethanol, xylenes, and paraffin themselves may extract key and important lipids/lipophilic compounds, or even lipoproteins in the structure of the retina. This potential extraction and elimination of key compounds may confound these studies as they as on a lipodystrophy.

Probably the authors ought to perform basically the same analyses as in Figure 6 and elsewhere, but with "frozen sections" that have never been exposed to any organic solvents that could extract the most important factors.

Minor points:

lines 5 and 6, Missing institutional affiliations for Tang and Li, and an extra "and" between Tang and Li.

lines 792-4. Unclear and please rephrase: "After removing the clot, the samples were centrifuged at 1000-x g for 10 minutes and serum was transferred into new microcentrifuge tubes for..."

It would seem more likely that the authors meant that the clot was separated from serum by the process of centrifugation, not before centrifugation.

Reviewer #2 (Remarks to the Author):

Age-related macular degeneration (AMD) is an incurable retinal disease and a top cause for blindness

in seniors worldwide. Causes for this disorder are complex, solid evidence links it to the accumulation of the drusen/lipofuscin under the retina over long period of time, that contains oxidated proteins and lipids leading to inflammation, neovascularization and impaired vision. Efforts have been taken to screen genetic defects that might link to AMD. Yuksel et al. utilized fundus spot scale together with a forward genetics pipeline to identify new candidates and found Lipe mutation is related to the accumulation of yellow fundus spots, a potential target for retinal degeneration. This is an interesting manuscript demonstrating how newly developed genetic technologies are applied to identify new targets for ocular disease that should be qualified for publication in Communications Biology with minor revisions. I have following comments.

Major concerns

1. To my best knowledge, the yellow fundus spots represent accumulation of drusen/lipofuscin in AMD patients. If this is true, start with this description will make audience clear that the discovery of Lipe gene in association with retinal degeneration is rationale without too much doubt.
2. Microglia are resident immune cells in the central nervous system (CNS) and the retina is part of CNS, thereby should be mostly locate in the retina. Their translocation from the retina to the subretinal space and the RPE is likely due to the accumulation of the yellow fundus spots. In the studies, authors only checked Iba-1+ cells in the RPE/Choroid flat mounts; but did not investigate their changes in the retina. Why?

Minor comments

1. Lines 124 – 129, change all the numbers 1., 2., ...5. to (1), (2), ... (5).
2. Lines 290 – 291, please cite references here.
3. Line 411 – 413, can you explain a tittle bit more about whether the autofluorescent spots represent drusen-like stuff?
4. Line 537, what does ENU stand for?
5. Lines 556 – 558, can you change "REF" to "WT", "VAR" to "NULL"?
6. Line 601, please spell out "sgRNA".
7. Line 609 – 611, as Lipe^{-/-} male mice were viable, why not use them together with Lipe^{-/-} female to generate Lipe^{-/-} offspring?
8. Line 620, RPE flat mounts are actually the RPE/Choroid flat mounts.
9. Line 790 – 791, after mice are euthanized, how to collect the blood?

February 16, 2023

Dear Communications Biology Reviewers:

Thank you very much for your careful evaluation of our manuscript, your positive comments, and your helpful suggestions. As a result of these, we have added 8 new figures, added a total of 5 new panels to 3 of the original figures, and made minor modifications to another 4 of the original figures. The most important changes to the text are highlighted in blue. Please find below a point-by-point response to the reviewers' suggestions. We hope you will agree that the modifications significantly strengthen our manuscript and make it suitable for publication at Communications Biology.

Sincerely,

Rafael L. Ufret-Vincenty, MD

Reviewers' comments:

Reviewer #1:

Summary: The authors present a great series of experiments that found several loci tightly linked to the appearance of fundus spots in mouse eyes. They have "...identified 14 genes that, when mutated, led to fundus spot accumulation." Here they characterized the most prominent locus and showed that the causative gene in the locus is *Lipe*, and its protein, hormone sensitive lipase E (HSL), that converts cholesterol esters to free cholesterol. The connection to cholesterol is intriguing.

Thank you for these positive comments.

Questions and concerns:

1. There are a limited number of reports of lipodystrophy due to mutations in *LIPE*. There are apparently no reports of retinal or macular diseases in these patients.

There is one report of *LIPE* mutation-related lipodystrophy in which three cases are presented (PMID 33112291). All three patients show retinal pathology involving the macula and midperiphery. They show prominent accumulation of fundus spots on fundus photography, which appear to be hyperautofluorescent. This is similar to our findings in the *Lipe* KO mice. This indicates that *Lipe* is not only relevant to retinal physiology in mice, but also in humans. Furthermore, these patients show very significant changes at the RPE level that are similar to AMD, including focal areas of RPE atrophy and focal subRPE deposits. These changes may not cause significant changes in electrophysiology testing or visual complaints until they affect the fovea, yet every clinician agrees they are extremely important in leading to later blinding complications (Lines 563-572).

2. Description of each of the 14 mutations that cause appearance of excess fundus spots

Thank you for the interest. We appreciate the fact that by including the number of genes of interest we have identified in our preliminary work, we are inviting the question regarding more details. We apologize for creating this issue. We have now removed the mention of the number of novel genes identified since that information is not relevant to this manuscript. We have kept the information regarding the 6 known and proof-of-principle genes found by our screen, including the Table S1 describing them. Our goal is to explore further our forward genetics findings by generating CRISPR KO mice to confirm the predicted gene-phenotype associations and understand them better. As you would understand, revealing the genes prematurely presents two problems: 1. It is entirely possible that some of those associations will not be confirmed in the CRISPR mice. 2. We have invested a huge amount of time, effort, and funds generating this preliminary data with the goal of exploring it further, so revealing it prematurely would compromise our future work.

3. Outside the obvious phenotype of many conspicuous spots on the fundus, other phenotypes (ONL thickness; ERGs, and so on) seem mild (though significantly decreased by statistical test, but still 90%+ of normal wildtype responses).

Thank you for bringing up this discussion point. The relatively mild impact on total retinal thickness is precisely one of the reasons we chose to pursue this gene in the first place. Our point is that our unique pipeline allows us to identify gene-phenotype associations that would not be pursued by others. Our mutagenesis and screening pipeline entails determination of all mutations and their zygosity within a pedigree prior to phenotypic screening. In traditional forward genetic screens, there is a huge amount of effort involved in pursuing any potential “hits” because many of them utilize outcross/intercross to a mapping strain in combination with exome sequencing to retrospectively identify causative mutations, a process that is time-consuming. This discourages the pursuit of any associations that do not lead to severe pathology. However, as we know, most human diseases do not involve extreme scenarios in terms of pathological changes, particularly in early stages. Also, a large number of retinal diseases are multifactorial and involve the combination of several “insults”. Thus, understanding genes that are relevant to retinal physiology is essential, even if mutations in those genes do not lead to total retinal collapse or blindness. Upon generating the CRISPR mutant lines, we were actually quite surprised to find that the *Lipe* mutation did indeed result in not only striking changes in fundus spots and retinal microglial activation, but also definite and progressive thinning of the outer nuclear layer and outer retina. Furthermore, we should point out that while some parameters only decreased by about 10% (e.g. total retinal thickness), other parameters decreased by close to 20% (e.g. outer nuclear layer thickness on both OCT and # of nuclear layers) and even 30% (ERG a-wave). Finally, we have now added data (Figure 9F) showing a significant impact on visual function using a separate parameter beyond ERG... optokinetic responses. The combination of these findings indicates that the bulk of the damage caused by HSL deficiency affects the photoreceptors. This magnitude of impact to the photoreceptors should probably be considered moderate, not

mild. This should be considered strong evidence that normal LIPE functioning is essential (non-redundant) to retinal homeostasis. These points are now included in several parts of the discussion (Lines 81-86, 347-351, 581-595)

4. Effects on rods versus cones are not clearly distinguished.

This is an excellent point. We have now performed photopic ERG (Suppl. Fig. S14) and confirmed that the effects on retinal function are statistically significant only for scotopic ERG, suggesting that LIPE deficits appear to affect mostly rods. We now include this in the results section (Lines 356-363).

5. Given these mild phenotypes even at relatively old age, what does this mean in terms of functional vision or disease etiology? The authors need to explain this better.

This is an important issue that we now address more with new data and additional discussion. Lipe mutations do not result in a blinding retinal dystrophy, which we do not suggest or imply in our manuscript. Our manuscript shows however, that Lipe does play an essential role in retinal homeostasis and that its absence leads to thinning of the outer nuclear layer as demonstrated by OCT and histology (now also using Cryo sections in the new Figure S12). Moreover, we demonstrate that the effect on visual function is also substantial. Lipe^{-/-} mice not only develop a statistically significant reduction in ERG signals, but we now include OptoMotry data showing a significant reduction in visual function as measured by OMRs (new figure panel 9F; p=0.006). The fact that deficits are not severe, do not mean that they are not relevant. First, we know that quality of vision is much more complex than what tests like visual acuity and electrophysiology can account for. More importantly, our point is that the retina is a very complex tissue that requires an exquisite orchestration of physiologic processes for proper function and that in many multifactorial diseases an alteration of multiple processes modulates disease severity. In our current work we demonstrate that Lipe plays an important and irreplaceable homeostatic role in the retina. It will of course, be of interest to study in the future if this process is involved in common multifactorial diseases like macular degeneration or diabetic retinopathy in which microglial activation and lipid dyshomeostasis are thought to play a role. This discussion is now included in lines (Lines 347-351, 581-595).

6. Could spots be a good thing? IE, If they are indeed activated microglia, could these spots be good microglia that promote repair and healing, or are they "bad" microglia that destroy cells that might be curable? The presumption is that these spots are bad and are promoting or causing degeneration of normal tissue, but maybe not... Explain how so (or not) in the manuscript.

The reviewer brings up an important point that we also want to emphasize. We tried to address this in the second sentence of our introduction by stating that: ***“There is still significant debate on whether these cells are deleterious (by promoting inflammation)¹⁻⁷ or homeostatic (by removing debris)⁸. In fact, it is possible that under different circumstances they may be either^{1, 9-13}.*** We also bring the topic up in the discussion’s first paragraph where we state that: ***“Mouse models of retinal degeneration (e.g. rd1, rd7, rd8, and rd10 models) confirm many of these***

conclusions^{1, 4, 6, 7}, but make it clear that the role of microglia may also be homeostatic, depending on both stimuli and anatomical location within the retina^{12, 13}.” However, after reading the reviewer’s comments it became clear that we need to reiterate this near the end of the discussion. We have added the following: “Future studies will also involve depleting microglia in *Lipe*^{-/-} mice and some of the other models we are generating in order to determine the role of subretinal microglia in the different pathologic processes we are observing. Specifically, we want to determine if microglia are helping return retina to homeostasis by clearing excess photoreceptor debris or causing damage by increasing inflammatory mediators.” (Lines 612-616). Finally, as part of this revision we have made an effort to better characterize the retinal and subretinal microglia in *Lipe*^{-/-} mice (see new supplemental Figures S5 and S6).

7. Some additional experimentation might be warranted, and possible additional testing that might bring out more dramatic and clearer phenotypes that might be more informative.

As mentioned above in response to Question #3, while total retinal thickness was only reduced by about 10%, ONL thickness was reduced by about 20% (both on OCT and in terms of the # of nuclear layers), and a-wave signal on ERG was reduced by about 30%. Still as described below, we have now added data on a measure of visual function (spatial frequency threshold on the OptoMotry system; new figure panel 9F) which confirms a statistically significant impact of the *Lipe* deficiency on visual function. Finally, the findings in the *Lipe*^{-/-} mice with regards to fundus spot accumulation and subretinal microglial accumulation are nothing short of dramatic. We will go over some of these findings in the answers below (answers #9 and #10). (Lines 338-343, 347-351).

8. ERG: Is there a difference in vision at a spot versus anywhere else?

Unfortunately, we do not have focal ERG capabilities at this time. However, we would not expect focal or regional differences in these mice, since we see the pathology evenly distributed across the retina.

9. More ERG analyses: c-waves and other electrophysiology tests

While we see some changes in the RPE of *Lipe* KO mice, these seem to be mostly due to increased phagocytosis of photoreceptor debris (Fig. 8). More detailed analysis of the EM images is now included (Fig. S13). It shows that there is no significant difference in the thickness of either the RPE or the basal infoldings in *Lipe* KO mice. There is possibly a mild increase in Bruch’s membrane thickness (which is seen in some human retinal degenerations), but this will need further exploration. Given these mild RPE changes, we were not expecting significant differences in c-waves on ERG. We now recorded scotopic c-wave ERG responses in *Lipe* KO vs WT mice (Fig. S14A) and confirm that no significant differences are detected.

In order to further characterize the functional impact of *Lipe* deficiency we now added several additional ERG tests. To address the issue of cones vs. rods, we have now tested photopic ERG responses (a-wave and b-wave, new Fig. S14B and S14C). As mentioned above, these are not different in *Lipe* KO mice, suggesting that most of the

impact in photoreceptors is involving rods rather than cones. Moreover, while we had already noticed on OCT that the inner retina of *Lipe* KO mice did not suffer any thinning, we also added oscillatory potential testing, which is known to correlate the most to inner retinal function. The results showed normal signals in *Lipe* KO mice (Fig. S14D), confirming that the inner retina is functioning well in these mice.

10. Other visual behavioral testing

Thank you for the advice. We decided to use the OptoMotry system to test the visual function of *Lipe*^{-/-} mice compared to *Lipe*^{+/+}. This system uses a rotating vertical sine wave grating to test the optokinetic reflex of mice, and is a well-recognized measure of visual function in mice. We found that *Lipe*^{-/-} mice have a significant reduction in the spatial frequency threshold when compared to *Lipe*^{+/+} (new figure panel 19F; p=0.006). This exciting finding establishes that the anatomic retinal abnormalities seen in *Lipe*^{-/-} mice have indeed a functional impact on vision. We agree with the reviewer that demonstrating that hormone sensitive lipase is essential to visual function using two completely different techniques (OMRs and ERG) significantly strengthens our conclusions.

11: Eye size: Are these mice myopic? Hyperopic? Could the spots and the slight thinning of the retina be functions of stretched retina?

Based on ocular measurements, the anteroposterior size of *Lipe*^{-/-} eyes is the same as that of *Lipe*^{+/+} mice. Thus, they should not have lacquer cracks or other myopic retinopathy changes that could explain the fundus spots. Also, this means that eye stretching is not the explanation for the significant thinning of the retina we observe in *Lipe*^{-/-} mice. We have added this data as Supplemental Figure S10G and included in the results section.

The thinning of the retina seems to be mostly affecting the outer neural retina. The decrease in the number of layers of nuclei in the ONL (which represents the nuclei of photoreceptors) and the thinning of the outer retina indicate that *Lipe*^{-/-} mice suffer from photoreceptor cell loss. In contrast their inner retina appears preserved (the ganglion cell complex thickness is not decreased).

12. Overloading the diet with specific lipids might provide a more distinctive functional phenotype.

This is indeed a very interesting experiment that we are hoping to do with enough funding. These experiments would last 9-12 months and would involve an extensive array of tests, including imaging, but also histological, scRNA sequencing, etc. We plan to submit a grant proposal using the work in this manuscript as preliminary data to justify the need for support to perform these studies.

13. What are the spots? While there seems to be evidence of co-registration between fundus spots and activated microglia, is there a perfect one-to-one relationship? Are activated microglia the cause of the fundus spots? Maybe some of the spots are activated astrocytes, Muller cells, invading macrophages, defective RPE cells (or other defective cell types, for example, a dying cone photoreceptor) that are migrating?

This is a very intriguing issue that many have tried to approach in other models in which fundus spot accumulations develop. So far, in all of these cases the conclusion has been that the spots represent microglia (see lines 170-171 and 466-467).

Currently we do not have a good way to do a one-to-one registration between the fundus spots and the activated microglia. Importantly, the fundus photos used to image the white/yellow spots are 2-dimensional representations of a spherical surface, while the flat mount used to image the Iba-1+ microglia is prepared by introducing radial cuts to flatten the spherical retina in a nonuniform manner. This information is now included in the discussion (Lines 472-475) Crossing the Lipe KO mice to CX3CR1-GFP mice would be an option. However, there are two issues with this approach: 1. While we think the fundus spots likely represent only the microglia that have reached the SUBRETINAL space, in CX3CR1-GFP mice all of the RETINAL microglia are fluorescent, making it challenging to distinguish the subretinal microglia or even establishing any sort of correlation... the microglia are just everywhere in the FAF image, 2. Crossing these mice while keeping the Lipe KO allele in the homozygous state and then aging them in order to allow the microglia to accumulate is likely a 1-year-long proposition. We are planning to include this experiment in future work.

Still, some of our findings in the Lipe KO mice point to the subretinal microglia as the reason for the fundus spots:

1. In Fig. S7, we made a strong effort to try to correlate the fundus spots to any findings on OCT. We superimposed the fundus photos on the IR images associated with the Spectralis OCT. This registration then allowed us to look with OCT at the exact location of the fundus spots. Using the vessels to confirm that we were on the correct location (red arrows) we were able to correlate the fundus spots to small hyperreflective spots located right above the RPE. Moreover, when the same eye was imaged 3 months apart, we could see how in a specific location in the fundus, there was no white/yellow spot on the fundus image (or hyperreflective spot on OCT) at 7m (Fig. S7B, white arrow) however, in the images obtained 3 months later we could see both a fundus spot in the fundus photo and a hyperreflective spot on OCT.
2. To provide further evidence that the fundus spots are not static, we have now **modified Fig. S7** to highlight the variety of changes we observe. We can see that while some new spots appear (yellow arrowhead in the magnified rectangle), some move and some disappear (green arrows). If we focus in the area shown by the white circle, there is a complete change in the distribution of the fundus spots. More examples of the changes happening even in a shorter time interval of 2 wks are shown in new figure panel Fig. S7C. These changes would be compatible with microglia, which are known to move.

14. "Do the same spots persist permanently, or do they change, move, or disappear and re-appear? Can the authors trace their movement over time?"

Please see the response to question #13 above and modified Figure S7, including new panel S7C.

15. Can the authors provide scale bars for the fundus images: How big are the spots compared to the spots by FAF, by SD-OCT, and by histology/confocal microscopy of Iba1 positive cells?

Scale bars of 200 μm are now included near the disc on the fundus photos in Fig. 2. The fundus spots vary significantly in size, but are in average 20-30 μm in size. In autofluorescence the spots are smaller (around 15-20 μm). However, it should be noted that if the autofluorescence is due to intracellular material, the size of the autofluorescent signal would be expected to be smaller in diameter than the entire cell. The subretinal spots that we show in Fig. S7 are around 40 μm in size, but we have chosen some of the larger spots, so this is within the range of the larger fundus spots (which can be up to 40-50 μm in size). While the entire size of subretinal microglia can be up to 40-50 μm , the cell bodies of the activated subretinal microglia is around 20-25 μm . All of these sizes are within range considering the fact that each of these modalities may highlight different properties of the cell (color in fundus photos, may not exactly correspond to autofluorescence, or to hyperreflectivity). This is now included in the discussion (Lines 478-489).

16. Are the spots associated with a dead or dying cell, clump of dying cells? TUNEL staining might be informative. Are these spots cellular or are they deposits of acellular debris or ECM? Oil slicks? Are they the beginnings of Drusen? Perhaps staining with some lipid stain (Oil red O, Nile red, etc.) be informative?

There does not appear to be any difference in TUNEL staining between the retina of *Lipe*^{+/+} and *Lipe*^{-/-} mice. Both of them show only occasional isolated apoptotic cells. There is no evidence of clumping of these and their number and size would not account for the fundus spots. This data is now shown in Suppl. Fig. S8. Also, we did not find any evidence of cellular deposits on light or electron microscopy. We also did Oil Red O staining on cryosections and did not find any evidence of lipid clumps in the retina or subretinal space in either KO or WT mice (data not shown).

17. If the spots were "funny" cones, then staining sections with cone opsins might be informative.

To address this question we stained retinal sections from *Lipe*^{-/-} mice and *Lipe*^{+/+} mice using antibodies against cone arrestin (cone-specific). We found no evidence of dysmorphic cones or clumps of cells (see new Suppl. Fig. S9). This is now included in the results section.

18. Other questions: In Figure 6, H&E staining, outer segments and inner segments of the *lipo* KO mouse appear less organized and more ragged than in WT. Is this consistent? If so, please report this finding. Since OS's are so dependent on proper creation of membranes, and *Lipe* likely governs this it may be worth discussing.

Thank you for this question. We looked carefully at all of our images and noted that there was some variation on the appearance of the photoreceptor outer and inner segments. We could not say with confidence that there was a consistent change in the *Lipo* KO mice. We have chosen a more representative section now for this figure.

19. Slight confounding factor: H&E's are generally done on paraffin embedded tissue (de-hydrated, re-hydrated). It appears that the authors fixed the eyes via a freeze-substitution technique (maybe-

probably -80 C Methanol: Acetic acid, 97:3; followed by ethanol.) The methanol, ethanol, xylenes, and paraffin themselves may extract key and important lipids/lipophilic compounds, or even lipoproteins in the structure of the retina. This potential extraction and elimination of key compounds may confound these studies as they as on a lipodystrophy. Probably the authors ought to perform basically the same analyses as in Figure 6 and elsewhere, but with "frozen sections" that have never been exposed to any organic solvents that could extract the most important factors.

This is an interesting point. However, there are several reasons that suggest that the freeze substitution method is optimal for the purpose of measuring ONL thickness:

1. While the freeze substitution protocol may extract some lipids, this is more likely to result in vacuolization of high-lipid content tissues in EM images, but not really thinning of the tissue.
2. The Outer Nuclear Layer is mainly composed of the cell bodies of the photoreceptors. Any potential impact of the lipid extraction should be maximal in the lipid-rich photoreceptor outer segments and minimal in the outer nuclear layer.
3. Freeze-substitution is based on rapid freezing of tissues followed by solution ("substitution") of ice at temperatures well below 0°C. This not only allows for the immobilization and stabilization of structures before the dehydration process, but also minimizes expansion/contraction artifacts and thus maximizes the preservation of tissue structure. In other words, the freeze substitution protocol leads to better anatomical preservation of all layers of the retina, which is essential for measurement purposes. Cryo sections introduce artifacts that can change measurements by causing tissue splits or artifactual cell-cell separations.
4. The counting of ONL nuclei is a surrogate measure of ONL thickness which is not affected by lipid extraction and still confirmed our finding of ONL thinning.
5. The ONL measurements obtained using our freeze substitution specimens are remarkably similar to those obtained in vivo by optical coherence tomography.

Still, we have now included confirmation of the ONL thinning on Cryo specimens (Supplemental Figure S12). While there is an artifactual increase in thickness of about 10 microns in both *Lipe*^{+/+} and *Lipe*^{-/-} specimens compared to OCT and freeze-substitution measurements, there is still a significant decrease in ONL thickness in *Lipe*^{-/-} compared to *Lipe*^{+/+} eyes.

20. Other histology issues

To better understand the microglial cells in *Lipe* deficient mice, we prepared retinal flat mounts and additional RPE flat mounts and imaged them with confocal microscopy after immunohistochemistry (Fig. 4). Interestingly, retinal flat mounts demonstrated that Iba1+ cells seen in the inner plexiform layer and outer plexiform layer of the retina have a similar morphology and distribution in *Lipe*^{-/-} mice compared to *Lipe*^{+/+} (Figure 4B vs 4A, and 4E vs 4D). These cells do not stain for CD16 (Fig. 5C and 5F). Meanwhile, cells in the subretinal space (stuck to the surface of RPE cells in the RPE flat mounts) are seen in *Lipe*^{-/-} (4H and 4K), but rarely in *Lipe*^{+/+} mice (4G and 4J), and they do stain for CD16 (4I and 4L). Moreover, both the retinal and the subretinal immune cells are TMEM119⁺ suggesting that they are microglia rather than infiltrating

monocyte/macrophages (Suppl. Fig. S4). In fact, flat mounts stained for F4/80 and TMEM119 (Suppl. Fig. S6) showed that the F4/80-stained cells (F4/80 recognizes both microglia and macrophages) are also TMEM119⁺ (specific marker for microglia), but CCR2 negative (CCR2 is a marker for macrophages).

21. Lines 5 and 6, Missing institutional affiliations for Tang and Li, and an extra "and" between Tang and Li.

Thank you for noticing these issues. They have been corrected.

22. Lines 792-4. Unclear and please rephrase: "After removing the clot, the samples were centrifuged at 1000-x g for 10 minutes and serum was transferred into new microcentrifuge tubes for...". It would seem more likely that the authors meant that the clot was separated from serum by the process of centrifugation, not before centrifugation.

Thank you for noticing this issue. It has been corrected.

Reviewer #2

This is an interesting manuscript demonstrating how newly developed genetic technologies are applied to identify new targets for ocular disease that should be qualified for publication in Communications Biology with minor revisions.

Thank you very much for these strongly supportive statements.

Questions and concerns:

1. To my best knowledge, the yellow fundus spots represent accumulation of drusen/lipofuscin in AMD patients. If this is true, start with this description will make audience clear that the discovery of Lipe gene in association with retinal degeneration is rationale without too much doubt.

Thank you for this comment. While yellow fundus spots in humans often represent drusen, in mice there are no good models of drusen. Many believe that this may partly be due to the much shorter life span, but the reason is not well understood. We and others have shown that these yellow fundus spots seem to correlate with the presence of activated subretinal microglia/macrophages. Please see the responses above, particularly #13, 15 and 21.

2. Microglia are resident immune cells in the central nervous system (CNS) and the retina is part of CNS, thereby should be mostly locate in the retina. Their translocation from the retina to the subretinal space and the RPE is likely due to the accumulation of the yellow fundus spots. In the studies, authors only checked Iba-1+ cells in the RPE/Choroid flat mounts; but did not investigate their changes in the retina. Why?

Thank you for this suggestion. We agree that it would be appropriate to look at retinal microglia, and have now added two new figures (Fig. 4 and Fig. S5) in which we stain them with Iba and TMEM119. The retinal flat mounts demonstrated that Iba1⁺ cells seen in the inner plexiform layer and outer plexiform layer of the retina have a similar morphology and distribution in *Lipe*^{-/-} mice compared to *Lipe*^{+/+}, with small cell bodies and long branching extensions. Contrary to subretinal microglia, these cells do not stain for

CD16. Moreover, both the retinal and the subretinal immune cells are TMEM119⁺ suggesting that they are microglia rather than infiltrating monocyte/macrophages (Suppl. Fig. S5).

Regarding the microglia migration, it is thought that activated microglia in the retina migrate towards the insult leading to their activation. In other words, in glaucoma and optic nerve injury models they migrate towards the inner retina, while in retinal dystrophy/degeneration models they tend to migrate towards the outer retina. However, we do not think that the microglia are migrating towards the fundus spots, but that the activated microglia ARE the fundus spots. Please refer to responses #13, 15 and 21.

3. Lines 124 – 129, change all the numbers 1., 2., ...5. to (1), (2), ... (5).

This change was made, and it does improve clarity.

4. Lines 290 – 291, please cite references here.

The following references were added:

39. Lyu Y, Tschulakow AV, Schraermeyer U. Melanosomes degrade lipofuscin and precursors that are derived from photoreceptor membrane turnover in the retinal pigment epithelium—an explanation for the origin of the melanolipofuscin granule. *bioRxiv*. 2022. doi:10.1101/2022.02.16.480523.

40. Bermond, K., et al. Autofluorescent Granules of the Human Retinal Pigment Epithelium: Phenotypes, Intracellular Distribution, and Age-Related Topography. *Invest Ophthalmol Vis Sci*. 2020;61(5):35. PMID 32433758

41. Robison, W. G., Jr., Kuwabara, T., Cogan, D. G. Lysosomes and melanin granules of the retinal pigment epithelium in a mouse model of the Chediak-Higashi syndrome. *Invest Ophthalmol*. 1975;14(4):312-7. PMID 1123287

5. Line 411 – 413, can you explain a little bit more about whether the autofluorescent spots represent drusen-like stuff?

We should note that we do not find lesions resembling drusen in light microscopy or EM. This is not uncommon, as true drusen are extremely rare in mice (PMID 20206286). Moreover, in multiple mouse models that have been reported to have drusen, the suspicious lesions have later been found to represent bloated subretinal microglia (PMID 19578022, 18421223). While we do not have an easy mechanism to determine a 1:1 correlation of autofluorescent spots to activated microglia and thus cannot prove the nature of these autofluorescent spots, it is interesting that their presence and distribution appears to be similar to that of the yellow fundus spots (Fig. S4). Also, the yellow fundus spots appear to correlate to subretinal hyperreflective spots on OCT (Fig. S5) and to subretinal microglia on RPE flat mounts (Fig. 3).

6. Line 537, what does ENU stand for?

ENU stands for N-ethyl-N-nitrosourea. This has now been defined at first mention (Line 113-114) and also in the Animals section (Line 648).

7. Lines 556 – 558, can you change “REF” to “WT”, “VAR” to “NULL”?

In the setting of ENU-generated pedigrees we prefer to refer to mice as REF, HET and VAR to emphasize that these mice have many other mutations and that we are only referring to the zygosity of the specific allele in question. We have clarified this in lines 647-650.

8. Line 601, please spell out “sgRNA”.

It means single guide RNA. It is now included in Line 694.

9. Line 609 – 611, as *Lipe*^{-/-} male mice were viable, why not use them together with *Lipe*^{-/-} female to generate *Lipe*^{-/-} offspring?

Male *Lipe*^{-/-} mice are sterile, so heterozygous males need to be used for breeding. This information is now included in the manuscript.

10. Line 620, RPE flat mounts are actually the RPE/Choroid flat mounts.

That is correct. We have now clarified this point by defining the term RPE flat mount as RPE-choroid-scleral flat mounts (line 173).

11. Line 790 – 791, after mice are euthanized, how to collect the blood?

Thank you for noticing this. The description has now been corrected as below (line 939):
“Prior to euthanasia, 10-11 month old *Lipe*^{-/-} and *Lipe*^{+/+} mice (n=6, per group) were anesthetized, eyes were enucleated and blood was collected in 2 ml micro centrifuge tubes from the orbital sinus.”

1. Aredo, B., et al. Differences in the distribution, phenotype and gene expression of subretinal microglia/macrophages in C57BL/6N (*Crb1* rd8/rd8) versus C57BL6/J (*Crb1* wt/wt) mice. *J Neuroinflammation*. 2015;12:6. PMID 25588310
2. Ma, W., Zhao, L., Fontainhas, A. M., Fariss, R. N., Wong, W. T. Microglia in the mouse retina alter the structure and function of retinal pigmented epithelial cells: a potential cellular interaction relevant to AMD. *PLoS One*. 2009;4(11):e7945. PMID 19936204
3. Madeira, M. H., Rashid, K., Ambrosio, A. F., Santiago, A. R., Langmann, T. Blockade of microglial adenosine A2A receptor impacts inflammatory mechanisms, reduces ARPE-19 cell dysfunction and prevents photoreceptor loss in vitro. *Sci Rep*. 2018;8(1):2272. PMID 29396515
4. Narayan, D. S., Ao, J., Wood, J. P. M., Casson, R. J., Chidlow, G. Spatio-temporal characterization of S- and M/L-cone degeneration in the Rd1 mouse model of retinitis pigmentosa. *BMC Neurosci*. 2019;20(1):46. PMID 31481030
5. Nebel, C., Aslanidis, A., Rashid, K., Langmann, T. Activated microglia trigger inflammasome activation and lysosomal destabilization in human RPE cells. *Biochem Biophys Res Commun*. 2017;484(3):681-6. PMID 28159556

6. Wang, N. K., et al. Origin of fundus hyperautofluorescent spots and their role in retinal degeneration in a mouse model of Goldmann-Favre syndrome. *Dis Model Mech.* 2013;6(5):1113-22. PMID 23828046
7. Zhao, L., et al. Microglial phagocytosis of living photoreceptors contributes to inherited retinal degeneration. *EMBO Mol Med.* 2015;7(9):1179-97. PMID 26139610
8. Ng, T. F., Streilein, J. W. Light-induced migration of retinal microglia into the subretinal space. *Invest Ophthalmol Vis Sci.* 2001;42(13):3301-10. PMID 11726637
9. Ambati, J., et al. An animal model of age-related macular degeneration in senescent Ccl-2- or Ccr-2-deficient mice. *Nat Med.* 2003;9(11):1390-7. PMID 14566334
10. Apte, R. S., Richter, J., Herndon, J., Ferguson, T. A. Macrophages inhibit neovascularization in a murine model of age-related macular degeneration. *PLoS Med.* 2006;3(8):e310. PMID 16903779
11. Kelly, J., Ali Khan, A., Yin, J., Ferguson, T. A., Apte, R. S. Senescence regulates macrophage activation and angiogenic fate at sites of tissue injury in mice. *J Clin Invest.* 2007;117(11):3421-6. PMID 17975672
12. O'Koren, E. G., et al. Microglial Function Is Distinct in Different Anatomical Locations during Retinal Homeostasis and Degeneration. *Immunity.* 2019;50(3):723-37 e7. PMID 30850344
13. Silverman, S. M., Ma, W., Wang, X., Zhao, L., Wong, W. T. C3- and CR3-dependent microglial clearance protects photoreceptors in retinitis pigmentosa. *J Exp Med.* 2019;216(8):1925-43. PMID 31209071

REVIEWERS' COMMENTS:

Reviewer #1 (Remarks to the Author):

The authors have thoroughly addressed my concerns. Nice job.

Reviewer #2 (Remarks to the Author):

I have read through the author's rebuttal letter and carefully checked their responses to the major concerns I raised in the review. I am satisfied with their responses and agree for the publication of their manuscript in Communications Biology.